# Fluorescence to measure light intensity

**Aliénor Lahlou** [1,2] ✉, **Hessam Sepasi Tehrani**[1], **Ian Coghill** [1], **Yuriy Shpinov**[1], **Mrinal Mandal**[1], **Marie-Aude Plamont** [1], **Isabelle Aujard**[1], **Yuxi Niu** [3], **Ladislav Nedbal**[3,4], **Dusan Lazár** [4], **Pierre Mahou** [5], **Willy Supatto** [5], **Emmanuel Beaurepaire** [5], **Isabelle Eisenmann**[6,7], **Nicolas Desprat** [6,7], **Vincent Croquette**[6,7], **Raphaël Jeanneret** [6,7], **Thomas Le Saux** [1] ✉ & **Ludovic Jullien** [1] ✉

Despite the need for quantitative measurements of light intensity across many scientific disciplines, existing technologies for measuring light dose at the sample of a fluorescence microscope cannot simultaneously retrieve light intensity along with spatial distribution over a wide range of wavelengths and intensities. To address this limitation, we developed two rapid and straightforward protocols that use organic dyes and fluorescent proteins as actinometers. The first protocol relies on molecular systems whose fluorescence intensity decays and/or rises in a monoexponential fashion when constant light is applied. The second protocol relies on a broad-absorbing photochemically inert fluorophore to back-calculate the light intensity from one wavelength to another. As a demonstration of their use, the protocols are applied to quantitatively characterize the spatial distribution of light of various fluorescence imaging systems, and to calibrate illumination of commercially available instruments and light sources.

Involved in key mechanisms of living systems (for example, photosynthesis, vision), photochemistry has found multiple applications at micro- and/or macro-scales from producing molecules[1] to designing medical protocols (for example, photodynamic therapy[2]). In bioimaging, optical microscopists balance light intensity to get optimal signals without phototoxicity. In optogenetics, biologists use photons for triggering physiological processes[3]. In photocatalysis[4], chemists exploit photons for driving the synergetic action of light-absorbing and metallic catalysts. Nowadays, a vast community of biologists, chemists, engineers and physicists are concerned with delivering precise numbers of photons.

Illumination systems require accurate quantitative characterization to ensure reproducibility as well as to enable a fair comparison of results obtained by various groups[5], or to rationally choose parameters such as duration of light application for delivering the right number of photons to a sample. Here we address light intensity, more precisely irradiance, which is a surfacic power (W m$^{-2}$), alternatively known as photon flux density in units of mol m$^{-2}$ s$^{-1}$ (or E m$^{-2}$ s$^{-1}$; Supplementary Note 4) that we will use in the following as it is wavelength independent.

Several tools are helpful to measure light intensity[6,7]. Light meters provide a fast response over a wide range of wavelengths and intensities[8,9]. However, their detectors integrate light over their surface and do not yield any information on the spatial distribution of light. A further measurement of the area of the illuminated surface is required to retrieve the light intensity. Fluorescent microscope slides deliver an image read-out of the spatial profile of illumination in imaging systems. Yet, the latter is affected by optical aberrations and the detection efficiency. Therefore, it cannot be relied on for retrieving accurate spatial information. Moreover, it does not give access to absolute light intensity. Eventually, light meters and fluorescent microscope slides sense

[1]PASTEUR, Department of Chemistry, École Normale Supérieure, PSL University, Sorbonne University, CNRS, Paris, France. [2]Sony Computer Science Laboratories, Paris, France. [3]Institute of Bio- and Geosciences/Plant Sciences, Forschungszentrum Jülich, Jülich, Germany. [4]Department of Biophysics, Faculty of Science, Palacký University, Olomouc, Czech Republic. [5]Laboratory for Optics and Biosciences, Ecole Polytechnique, CNRS, INSERM, IP Paris, Palaiseau, France. [6]Laboratory of Physics of the École Normale Supérieure, University of PSL, CNRS, Sorbonne University, University of Paris City, Paris, France. [7]Institute of Biology of ENS (IBENS), École Normale Supérieure, CNRS, INSERM, University of PSL, Paris, France. ✉e-mail: Alienor.Lahlou@sony.com; Thomas.Lesaux@ens.psl.eu; Ludovic.Jullien@ens.psl.eu

light at sample surfaces only and experience geometrical constraints from relying on large and rigid sensing elements.

Actinometers provide an alternative approach. Light intensity is directly retrieved from following the time course of their reaction extent on constant illumination[10]. This approach can further yield spatial information when used with an imaging system. Moreover, since they take the form of liquid solutions, actinometers can measure light in samples of various sizes and geometries. However, most established and new actinometers have relied on absorbance to report on reaction extent[10–12]: an observable that is not very sensitive and is not easily accessible in imaging systems. Furthermore, they are generally restricted for the ranges of wavelengths and light intensities.

Fluorescence is a more sensitive observable than absorbance[13] and is accessible to imaging systems. Harnessing fluorophore photobleaching has been proposed for quantitative measurement of light intensity[14]. However, the photobleaching kinetics can be complex and exhibit environmental dependence, and therefore the use of photobleaching kinetics is limited to situations of high light levels or long time periods since most fluorophores are strongly resistant to photobleaching.

Here, we demonstrate the use of synthetic and genetically encoded fluorescent photoactivatable systems that we previously reported[15–19] as actinometers, whereby fluorescence is used for reporting the extent of the photoconversion reaction. Such conversions proceed much more rapidly than photobleaching, and thus these systems can be used in weak-light situations (or short time periods). To allow for measurement over a wider range of wavelengths, we complement our proposed method with a broad-absorbing photochemically inert fluorophore that enables light intensity at one wavelength to be used to calculate light intensity at a second wavelength. In this Article, we report on the relevant features of these systems for measuring light intensity and demonstrate their use in characterizing illumination systems at both microscopic and macroscopic scales with samples of various sizes and geometries.

## Results

### The first protocol to measure light intensity

The protocol to measure light intensity (Fig. 1a) exploits molecular actinometers, which react on absorbing the excitation light to be characterized (Supplementary Note 5). The absorbance of their solution is adjusted low enough to ensure that light intensity is essentially constant along the optical path (below 0.15, an easily met threshold). The protocol begins with the sudden exposure of the actinometer solution to illumination, set at the level of light intensity $I$ to be measured. One subsequently collects the time evolution of the fluorescence signal, which reports on the actinometer photoconversion extent. It is processed by the fitting of a monoexponential curve, to enable the retrieval of the associated characteristic time $\tau$, which evaluates the time scale of the actinometer photoconversion. In an appropriate range of light intensity, $I$ is equal to the inverse of the product $\sigma\tau$ where $\sigma$ is the photoconversion cross section (a measure of the molecular surface leading to the actinometer photoconversion after light absorption) (Fig. 1a) and it is measured within a 20% achievable uncertainty (Supplementary Note 7). Where photoconversion occurs rapidly, on a time scale where molecular motion is minimal, it is possible to retrieve a map of the spatial distribution of light intensity. However, if the molecules can visit the whole irradiated area at the time scale of the actinometer photoconversion, only mean light intensity values can be obtained[16].

The choice of the reported actinometers has been guided by several considerations. First, we wanted to illustrate two mechanisms enabling fluorescence to be used as a reporter of their photoconversion. In the first one, the actinometer and/or its photoproducts are intrinsically fluorescent: the fluorescence intensity changes in accordance with the actinometer photoconversion and the actinometer alone is sufficient to retrieve $I$. This mechanism is simple but its implementation relies

on specific features, which led us to identify appropriate candidates. Indeed, photoconversion and fluorescence emission are competitive deexcitation processes from an excited state. By contrast, when the actinometer and its photoproducts are nonfluorescent, the actinometer photoconversion only drives a change of absorbance. Here, a photochemically inert fluorophore can be added to report, through fluorescence, on the time evolution of the absorbance at the excitation wavelength by the inner filter effect (Supplementary Information Section 5.1.2). In this last mechanism, the absorbance is advantageously adjusted around 0.15 to increase the amplitude of the time variation of the fluorescence signal, and thus sensitivity. Then we wanted to make fluorescent actinometers accessible to different communities of end users. Hence, we report on easily synthesized chemicals for the chemists, whereas we propose to use proteins and photosynthetic organisms for end users with access to biological techniques. Eventually, the five reported actinometers cover the entire ultraviolet-visible light (UV-vis) wavelength range for measurement of light intensity (Fig. 1c and Supplementary Information Sections 7.4–7.6):

- Two actinometers for the UV-A wavelength range (relevant for decontamination of materials[20], evaluation of environmental radiation[21], photoactivation of many caged molecules in optogenetics[22], photocatalysis with metal complexes[4] and so on): (1) the dark (*E*)-3-(3,5-dibromo-2,4-dihydroxyphenyl) acrylic acid ethyl ester (Cin)[15,23], which irreversibly converts under illumination between 350 and 420 nm into the bright 6,8-dibromo-7-hydroxycoumarin fluorescing between 400 and 550 nm in Tris pH 7 buffer and (2) the dark $\alpha$-(4-diethylamino)phenyl)-*N*-phenylnitrone (Nit)[24], which irreversibly converts under illumination between 320 and 430 nm into the dark *N*-(*p*-dimethylaminophenyl)formanilide in ethanol[19]. The photochemically inert rhodamine B (RhB) emitting fluorescence between 550 and 650 nm is selected here for optimally reporting on Nit photoconversion by inner filter effect.

- One actinometer for the blue wavelength range (important in optogenetics for photoactivating opsins, FAD CRY, FAD BLUF and FMN LOV systems[3], or driving photosynthesis[25]). A bright reversibly photoswitchable fluorescent protein Dronpa-2 (or M159T)[26], contained within *Escherichia coli* or eucaryotic cells, or in buffered solution or polyacrylamide gel, emitting fluorescence between 500 and 600 nm, which reversibly converts into a dark photoisomerized state under illumination between 400 and 550 nm.

- One actinometer for the green to red wavelength range (important for photoactivating opsins or bilin PHY[3] in optogenetics, or driving photosynthesis[26]). In acetonitrile, the donor-acceptor Stenhouse dye DASA (sodium 4-(4-((2*Z*,4*E*)-2-hydroxy-5-(indolin-1-yl)penta-2,4-dien-1-ylidene)-3-methyl-5-oxo-4,5-dihydro-1H-pyra-zol-1-yl) benzenesulfonate)[17] emitting fluorescence extending up to 675 nm reversibly converts into a dark state under illumination between 530 and 670 nm.

- As the width of the absorption band of the preceding fluorescent actinometers is limited, which necessitates several of them covering the whole range of wavelengths, we eventually report on the last actinometer, the photosynthetic apparatus of algae, which can provide an estimate of light intensity for the entire visible range of wavelengths. In oxygenic photosynthetic organisms, a few percent of collected sunlight energy is released as fluorescence in the 650–800 nm range[27]. When exposed to constant light at sun-like intensity, the fluorescence of dark-adapted photosynthetic organisms rises in less than 1 s from a minimum to a maximum via intermediate steps[28,29]. The rate constant of the fastest step linearly depends on the light intensity[30,31]. Usefully, its value does not considerably depend on the type of photosynthetic organism[32].

Characterization and validation of the five above-mentioned actinometers as well as specific protocols for their use are reported

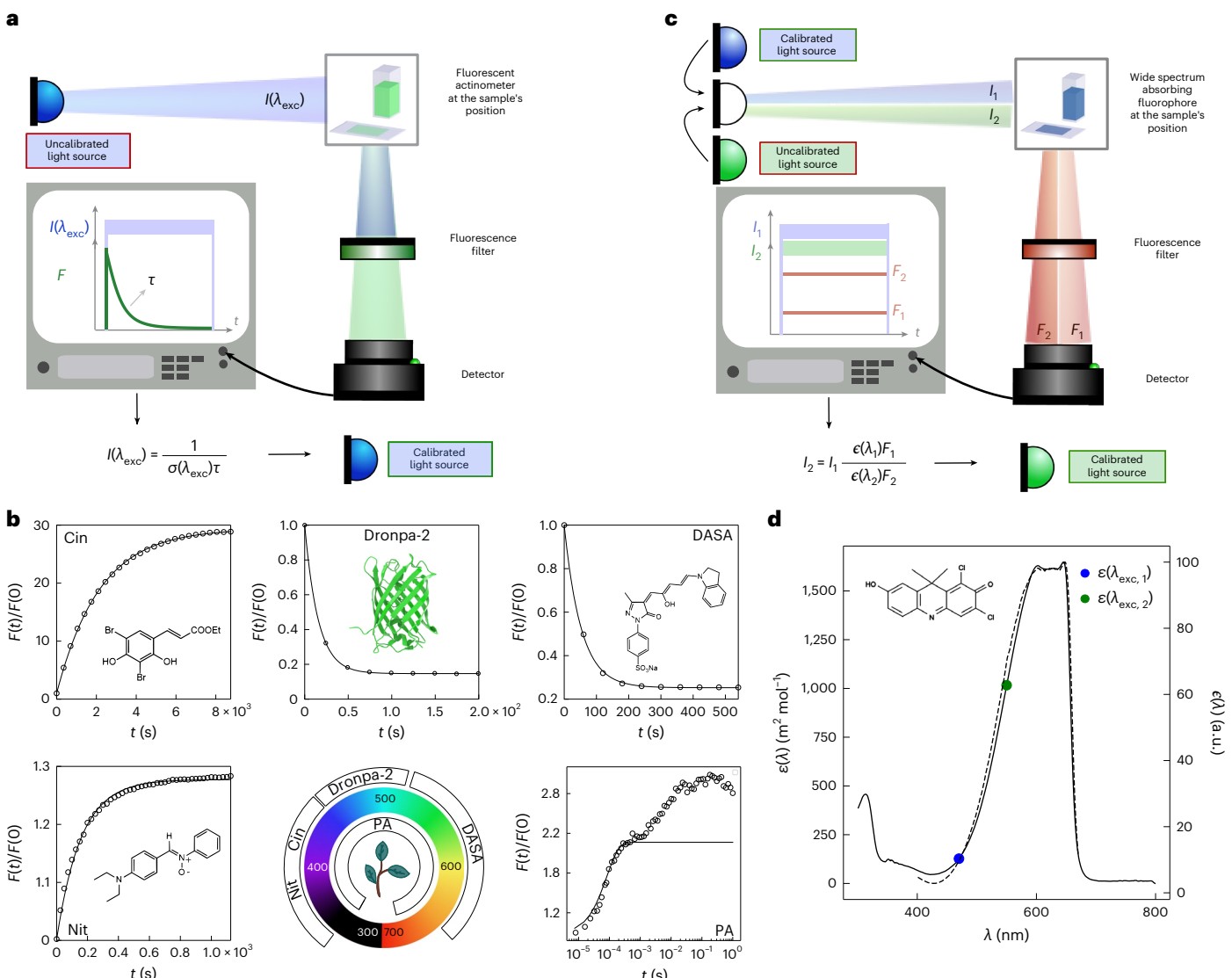

**Fig. 1 | Fluorescence reporting for retrieving light intensity. a**, First protocol with a fluorescent actinometer. A jump of constant light $I$ is applied onto the actinometer. The time evolution of its fluorescence signal $F$ is recorded and fit with a monoexponential curve to extract its characteristic time $\tau$. $I$ is retrieved from $\tau$ by using the photoconversion cross section $\sigma$ of the actinometer. **b**, Five fluorescent actinometers covering the UV-vis range in action. Monoexponential fit of the time evolution of the normalized fluorescence signal $F(t)/F(0)$ provides $\tau$ (Supplementary Table 3). **c**, Second protocol with a fluorophore to transfer information on light intensity from one wavelength to another. Lights at

wavelengths $\lambda_1$ (with intensity $I_1$, known) and $\lambda_2$ (with intensity $I_2$, to be measured) are successively applied onto the fluorophore and the associated fluorescence signals $F_1$ and $F_2$ are recorded at a same emission wavelength. $I_2$ is extracted from $F_1$ and $F_2$ by using $I_1$ and the tabulated fluorescence excitation spectrum $\varepsilon(\lambda)$ of the fluorophore. **d**, Absorption ($\varepsilon(\lambda)$; dotted line) and normalized fluorescence excitation ($\varepsilon(\lambda)$; solid line) spectra of DDAO. $\varepsilon(\lambda_1)$ and $\varepsilon(\lambda_2)$ indicated by blue and green disks, respectively, are used to retrieve $I(\lambda_2)$ in **c** (text and Supplementary Tables 1–3).

in the Supplementary Notes 2 and 7, respectively. Table 1 provides information to facilitate the selection of the most optimal actinometer to use for specific scenarios. The tabulated parameters will enable end users to reliably measure light intensity as long as they follow the reported measurement protocols. To further facilitate the use of these actinometers, we provide online access to the actinometer properties (https://chart-studio.plotly.com/~Alienor134/#/) and to codes and user-friendly applications to process the acquired data without specific installation (https://github.com/DreamRepo/light_calibration).

### The second protocol to measure light intensity

The second protocol to measure light intensity (Fig. 1c) is envisioned as a complementary tool to face the limited absorption bandwidth of

the fluorescent actinometers; it exploits a photochemically inert fluorophore exhibiting a broad absorption band to transfer information on light intensity from one wavelength, measured with a fluorescent actinometer as reported above, to another (Supplementary Note 5). Below a value of 0.15 for the absorbance of its solution, the fluorophore emits fluorescence at an intensity proportional to light intensity of the illumination system. By recording the fluorescence emitted when the fluorophore solution is exposed to light at a wavelength ($\lambda_1$) where the light intensity ($I_1$) is known, and a wavelength ($\lambda_2$) where the light intensity ($I_2$) is unknown, the unknown light intensity ($I_2$) can be determined (Fig. 1c).

As demonstrated in Supplementary Information Section 7.6, 7-hydroxy-9H-(1,3-dichloro-9,9-dimethylacridin-2-one) (DDAO) is a suitable light intensity-transferring fluorophore (Fig. 1d). It is

**Table 1 | Key parameters for choosing a fluorescent actinometer**

| Actinometer (availability)[a] | $\lambda_{exc}$ (nm) | $\lambda_{em}$ (nm) | $\sigma(\lambda_{exc})$ (±10%, m²mol⁻¹) | $(i(\lambda_{exc}))$ (E m⁻²s⁻¹(W m⁻²))[b] | $5\tau_{min}$ (s) |
|---|---|---|---|---|---|
| **Cin** (S) | 350 | | 940 | $(0–1.6)10^{-5}((0–5.5))$ | 332 |
| | 365 | 400–550 | 1,200 | $(0–1.4)10^{-5}((0–4.6))$ | 298 |
| | 380 | | 1,000 | $(0–1.7)10^{-5}((0–5.3))$ | 294 |
| | 405 | | 184 | $(0–5.0)10^{-5}((0–15))$ | 543 |
| | 420 | | 49 | $(0–1.3)10^{-4}((0–37))$ | 785 |
| **Nit** (S) | 365 | | 960 | $(0–11)10^{-4}((0–360))$ | 4.7 |
| | 380 | 550–650 | 1,200 | $(0–7.2)10^{-4}((0–230))$ | 5.8 |
| | 405 | | 1,100 | $(0–7.0)10^{-4}((0–200))$ | 6.5 |
| | 420 | | 850 | $(0–12)10^{-4}((0–340))$ | 4.9 |
| **Dronpa-2** (GE) | 445 | | 140 (192)[c] | $(3\times10^{-4}–18)((80–4.8\times10^{6}))$ | $2.0\times10^{-3}$ |
| | 480 | 500–600 | 198 (251)[c] | $(2\times10^{-4}–10)((50–2.5\times10^{6}))$ | $2.5\times10^{-3}$ |
| | 500 | | 128 (151)[c] | $(3\times10^{-4}–13)((72–1.0\times10^{6}))$ | $3.0\times10^{-3}$ |
| **DASA** (S) | 530 | | 255 | $(8–290)10^{-5}((18–660))$ | 6.8 |
| | 560 | 530–670 | 530 | $(4–150)10^{-5}((9–320))$ | 6.3 |
| | 600 | | 885 | $(2–72)10^{-5}((4–140))$ | 7.8 |
| | 632 | | 1,135 | $(2–60)10^{-5}((4–110))$ | 7.3 |
| | 650 | | 575 | $(3–140)10^{-5}((6–260))$ | 6.2 |
| **PA** (B) | 405 | | 2.0 (±0.4) $10^{6}$ | $(0–10^{-2})((0–3,000))$ | 1[d] |
| | 470 | 650–800 | 2.0 (±0.4) $10^{6}$ | $(0–10^{-2})((0–2,600))$ | 1[d] |
| | 650 | | 1.1 (±0.4) $10^{6}$ | $(0–10^{-2})((0–1,900))$ | 1[d] |

$\lambda_{exc}$, $(\lambda_{em})$, $\sigma(\lambda_{exc})$ and $(I(\lambda_{exc}))$, respectively, designate the excitation wavelength, the range of emission wavelengths, the cross section and the range of reliably measurable light intensity associated to the actinometer photoconversion at $\lambda_{exc}$. $5\tau_{min}$ is the minimum measurement duration at the highest measurable light intensity. See Supplementary Note 7. [a]S, easily synthesized; GE, genetically encoded; B, photosynthetic organism available for sale (chlamycollection.org) (Supplementary Note 1). [b]The conversion between the units mol of photon m⁻²s⁻¹, E m⁻²s⁻¹ and W m⁻² is given in Supplementary Note 4. [c]The first and second numbers provide the values to be used in the Dronpa-2 solution and in the Dronpa-2 labeled fixed cells, and in the Dronpa-2 labeled bacteria, respectively. [d]$5\tau_{min}$ is the time requested to record the whole photosynthetic apparatus fluorescence rise.

commercially available, absorbs light between 450 and 650 nm and emits fluorescence between 640 and 700 nm in neutral aqueous solutions[33,34], and its quantum yield of fluorescence does not depend on the excitation wavelength as evidenced by the similarity of its absorption and fluorescence excitation spectra. These features are particularly attractive for light calibration in the orange and red wavelength range where actinometers are scarce and often exhibit a poor quantum yield of fluorescence.

## Measurement of light intensity in fluorescence imaging systems

Accurate measurement of light intensity is important in many fluorescence bioimaging studies (for example, to limit phototoxicity on live biological samples[35], for quantitative analysis in long timelapse[36] or ratiometric[37] studies, or optimal conditions in single molecule localization[38] and dynamic contrast[16,39]). Hence, we first used fluorescent actinometers for measuring light intensity at the focal plane of multiple wide-field and light-scanning fluorescence imaging systems. Dronpa-2 has been used here for this.

We first implemented Dronpa-2 for wide-field epifluorescence microscopy. A Dronpa-2 aqueous solution sandwiched between two glass slides was imaged (Fig. 2a) and subjected to a light jump at 470 nm. Figure 2b shows the relevance of a monoexponential fit of the resulting temporal fluorescence decay over the field of view. We retrieved the map of the characteristic time $\tau$ at each pixel (Fig. 2c) and built the histogram of the $\tau$ values (Fig. 2d). The map (Fig. 2e) and the corresponding histogram (Fig. 2f) of light intensity at 470 nm were subsequently computed from using the photoconversion cross section of Dronpa-2 in Table 1. We further exploited a patterned illumination at 470 nm (Fig. 2g) and observed that the map of light intensity was obtained at lower spatial resolution when Dronpa-2 was in solution (Fig. 2h) than when it was embedded in a polyacrylamide gel (Fig. 2i), albeit with identical quantitative information (Extended Data Fig. 1). This result was anticipated from molecular diffusion occurring during Dronpa-2 photoconversion, which generates blurring (Supplementary Information Section 2.1.4).

For quantitative validation, Dronpa-2 was then implemented in fluorescence macroimaging on an original optical setup (Supplementary Information Section 7.3.2), whose illumination is not homogeneous but instead contains a gradient of light intensity across the field of view (Fig. 2j). The sandwiched Dronpa-2 solution was submitted to a light jump at 470 nm to extract the map of the characteristic time $\tau$, which was converted into the map of light intensity. This latter map was validated by favorable comparison of the direction of the linear gradient of light intensity, either experimentally observed (Fig. 2k) or computed from an optical simulation (Fig. 2l and Supplementary Information Section 8.1). The lines along the gradient direction differ from each other only by an angle of 3°.

A similar protocol was applied in confocal microscopy. A series of images of the Dronpa-2-labeled nucleus of a fixed U-2 OS cell were acquired in raster scanning mode with a pulsed laser at 488 nm (Fig. 3a). The dwell time was used to convert the observed fluorescence decay on the number of frames into a fluorescence drop over time. Figure 3b displays the average drop over a nucleus. It also shows that a Dronpa-2 aqueous solution sandwiched between two glass slides yields a similar kinetic signature on properly restricting analysis to a central portion of the overall image to limit the interference of molecular diffusion on the results (Supplementary Information Section 2.3.3). Maps of the characteristic time $\tau$ (Fig. 3c) and light intensity (Fig. 3e), and the corresponding histograms (Fig. 3d,f, respectively), were obtained. The mean light intensity retrieved was shown to be consistent with that calculated using the photon flux measured with a power meter, combined with area measurements of the waist of the laser beam, evaluated by raster image correlation spectroscopy[40] or imaging a fluorescent bead (Supplementary Information Section 8.2). The same series of experiments and validations were performed using a confocal microscope equipped with a continuous, rather than pulsed, laser (Extended Data Fig. 2 and Supplementary Information Sections 2.3.3 and 8.2). The results of which confirmed the ability of this protocol to retrieve light intensity with both modes of laser scanning.

We eventually performed measurements with Dronpa-2-labeled *E. coli* bacteria cells with and without a layer of 2% agarose gel between them and the imaging system (Fig. 3g). Light was applied with the deposited layer facing the objective, and then again after the sample

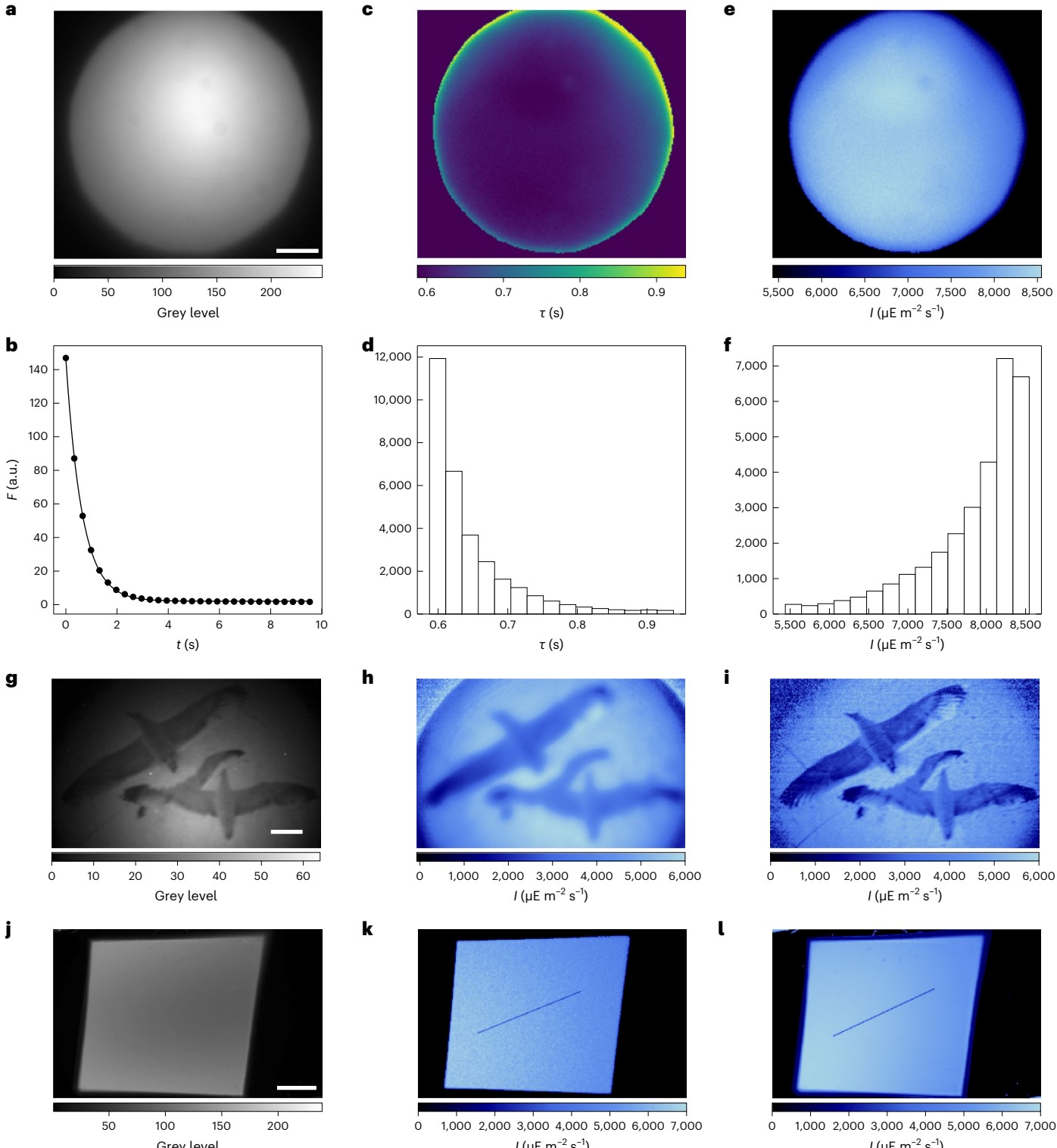

**Fig. 2 | Dronpa-2 for characterization of illumination in wide-field fluorescence imaging. a–i**, Epifluorescence microscopy. **a**, Initial image of the Dronpa-2 solution under homogeneous illumination. **b**, Time fluorescence response ($\tau = 0.63$ s), maps (**c**,**e**) and histograms (**d**,**f**) of the characteristic time $\tau$ (**c**,**d**) and light intensity (**e**,**f**) in the field of view. **g–i**, Initial image (**g**) and maps of light intensity (**h**,**i**) of Dronpa-2 in solution (**h**) or in polyacrylamide gel under patterned illumination. **j–l**, Macroscopic fluorescence imaging: initial image of the Dronpa-2 solution (**j**) and experimental (**k**) and simulated (**l**) maps of light intensity. The blue line shows the angle of the linear light gradient; the angle between the simulated gradient and the measured one is 3°. **a–f,h,j–l**, 10 μM Dronpa-2 solution or 19% polyacrylamide gel in Tris buffer pH 7.4 (50 mM Tris, 150 mM NaCl). Scale bars, 100 μm (**a,c,e,g,h**); 3 mm (**j–l**). $T = 293$ K. $\lambda_{exc} = 470$ nm; $\lambda_{em} = 550$ nm (text and Supplementary Tables 1 and 2). Independent repeats, more than 30 (**a,c,e**); 3 (**g–i**); 15 (**j,k**).

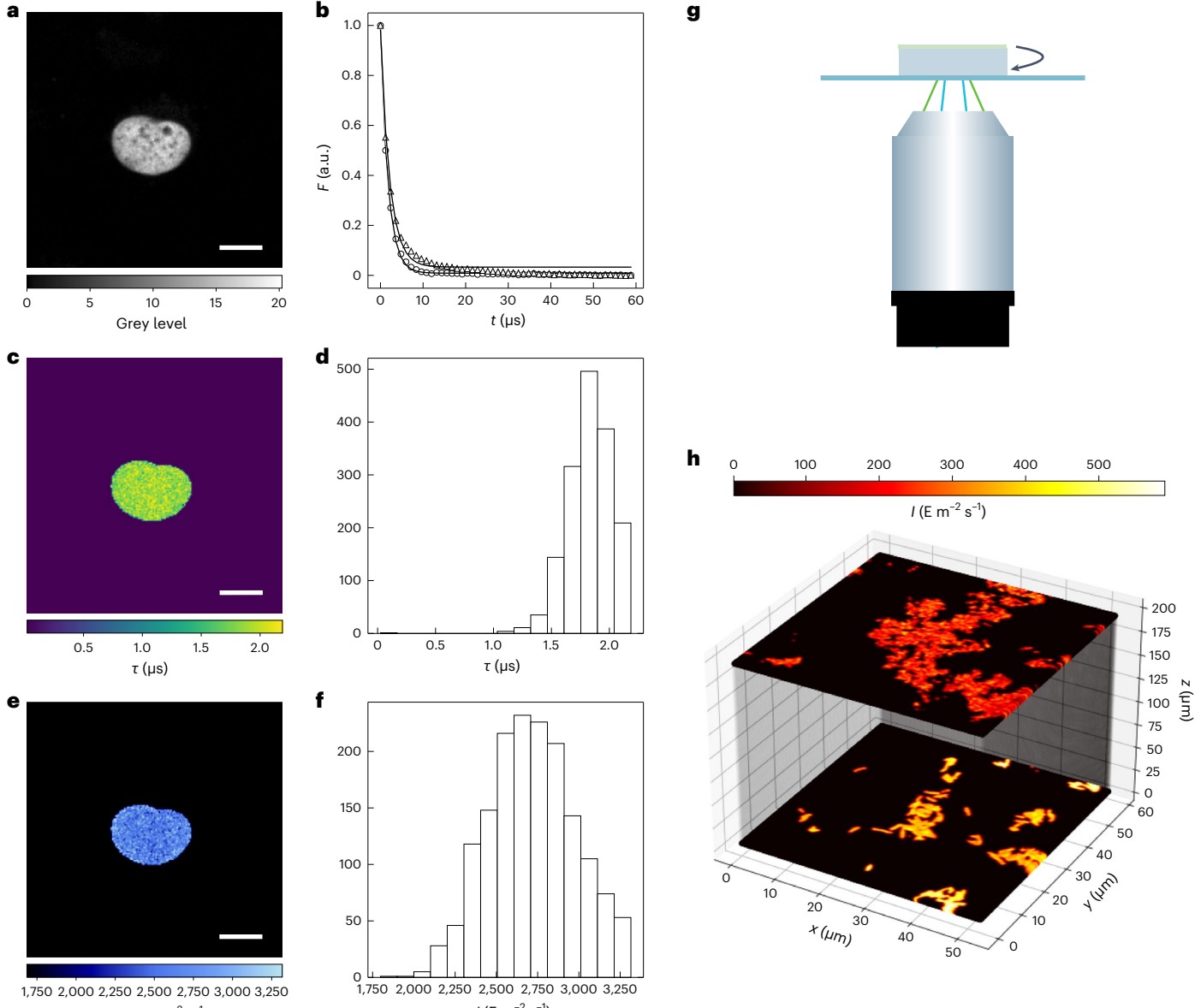

**Fig. 3 | Dronpa-2 for characterization of illumination in confocal microscopy equipped with a pulsed laser in the raster scanning mode. a–f**, Dronpa-2-labeled nucleus of a fixed U-2 OS cell. **a**, Initial image. **b**, Time evolution of the averaged fluorescence over the whole nucleus (circles, experimental data; solid line, monoexponential fit, $\tau = 1.9\,\mu s$). The corresponding evolution from a central portion of the overall image of a 10 μM Dronpa-2 solution sandwiched between two glass slides is shown with triangles ($\tau = 2.1\,\mu s$). **c–f**, Maps (**c,e**) and histograms (**d,f**) of the characteristic time $\tau$ (**c,d**) and light intensity (**e,f**) (Supplementary Table 3). **g,h**, Setup (**g**) and map of light intensity retrieved from Dronpa-2-labeled *E. coli* bacteria imaged at the surface or through a 2% agarose pad by changing the sample orientation (**h**). Solvent was Tris buffer pH 7.4 (50 mM Tris, 150 mM NaCl); *T* = 293 K. $\lambda_{exc} = 488\,nm$; 500 nm < $\lambda_{em}$ < 550 nm. Scale bar, 12 μm (**a,c,e**) (text and Supplementary Tables 1–3). Independent repeats, 4 (**a,c,e**); 3 (**h**).

was flipped, such that the light had to cross the gel before reaching the cells. The maps of light intensity for both orientations can be measured in bacteria cells even when buried behind the agarose gel (Fig. 3h), showing that this protocol can measure light intensity, not just at the surface, but in situ, deep within samples.

**Calibration of setting scales of light intensity.** Many optical instruments do not provide information on the absolute light intensity, but rather just the percentage of the maximum possible light that may change with time as the light source ages. Thus, as a second application, we applied the Dronpa-2 and photosynthetic apparatus actinometers for the calibration of the percentage scales of a confocal microscope and a fluorometer, respectively.

Using a Dronpa-2 aqueous solution sandwiched between glass slides, we measured the light intensity of a confocal microscope equipped with a pulsed laser at 488 nm at different percentages of the maximal laser power, using the protocol detailed above for confocal systems (Fig. 4a). Then we generated the same type of calibration graph (Fig. 4b) for a fluorometer by analyzing the photosynthetic apparatus fluorescence kinetics of a dark-acclimated suspension of microalgae exposed to light at 625 nm (Supplementary Information Sections 2.4.3 and 5.2). These experiments demonstrated that this protocol can allow end users to convert the percentage value indicated on the instrument into absolute light intensity, and to verify linearity in the range of settings investigated.

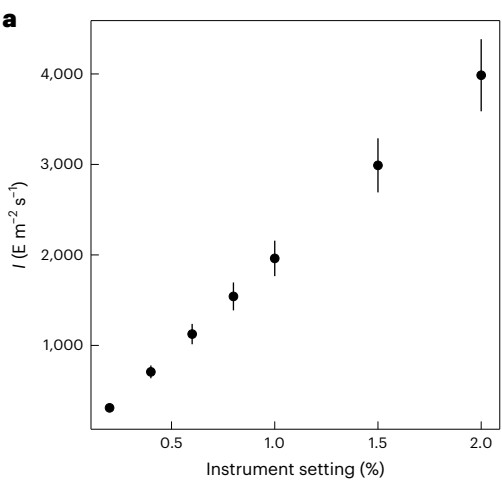

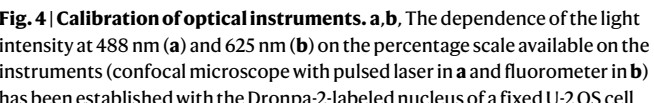

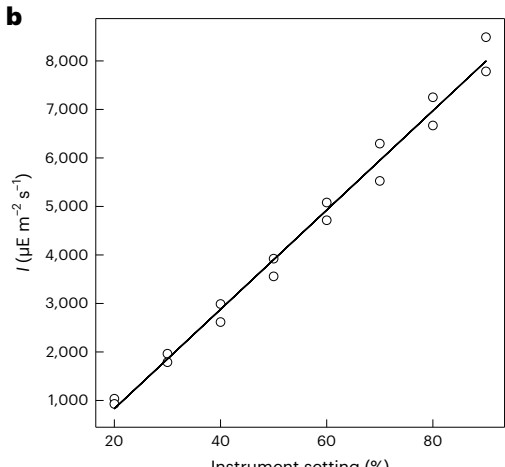

**Fig. 4 | Calibration of optical instruments. a,b**, The dependence of the light intensity at 488 nm (**a**) and 625 nm (**b**) on the percentage scale available on the instruments (confocal microscope with pulsed laser in **a** and fluorometer in **b**) has been established with the Dronpa-2-labeled nucleus of a fixed U-2 OS cell (mean value ± s.d., s.d. are propagated from the error on σ(488) (20 repeats)) and photosynthetic apparatus actinometers, respectively. *T* = 293 K (text and Supplementary Tables 1 and 2).

**Spectral measurement of light intensity.** Photons at different wavelengths drive photoconversion of actinometers to differing degrees. Hence, the spectral characteristics must be considered to obtain accurate measurements of light intensity for light sources, which do not emit at a single wavelength but rather over a spectrum of wavelengths. Accordingly, we extended our protocol to deliver spectral light intensity of nonmonochromatic light sources.

This extended protocol begins with matching the excitation spectrum of the actinometer with the emission spectrum of the light source made available by the manufacturer or measured with a spectrophotometer (Supplementary Information Section 7.2.2). After obtaining the time course of the fluorescence intensity as before, the retrieved characteristic time is used in conjunction with the integral of the action spectrum, the convolution of the normalized emission spectrum of the light source with the excitation spectrum of the actinometer, to quantitatively extract the spectral light intensity (Supplementary Information Section 2.5).

As preliminary application, we used Nit and photosynthetic apparatus to characterize the illumination from purple and red-orange light emitting diodes (LEDs), respectively (Extended Data Figs. 3 and 4). Then we turned to the more challenging task of measuring the spectral light intensity of a white LED by using DDAO to implement the protocol for transferring information on light intensity (Fig. 5a, Extended Data Fig. 5 and Supplementary Information Sections 2.2, 2.5 and 5.3). We measured the DDAO fluorescence intensity when illuminated by a blue LED previously calibrated with the Dronpa-2 actinometer, and by the white LED. The ratio of the fluorescence signals obtained was used, in combination with the known light intensity of the blue LED, to infer the light intensity of the white LED. The light intensity was then spectrally corrected by using the integral of the normalized action spectrum between the white LED and DDAO (Fig. 5b). This experiment was repeated to deliver the integrated light intensities for a range of LED current settings (Fig. 5c) and the scaled spectral light intensity of the white LED (Fig. 5d).

## Discussion

The measurement of the absolute flux of light at the sample (Working Group 1) and the assessment of the uniformity of illumination (Working Group 3) are key issues of the QUAREP-LiMi community (Quality Assessment and Reproducibility for Instruments and Images in Light Microscopy; https://quarep.org/) made accessible by fluorescent actinometers[6]. After a selection relying on the wavelength to investigate and access to chemical or biological facilities, end users should

implement the protocol illustrated in Fig. 1a and detailed in Supplementary Note 2 on using https://github.com/DreamRepo/light_calibration/releases/. Alternatively, they should use the commercially available DDAO to retrieve the light intensity of their desired light source after calibrating another light source with the actinometer they can access. Beyond wide-field fluorescence micro- and/or macroimaging and confocal microscopy explored here, our previous use of quantitative photoconversions suggests the fluorescent actinometers to be relevant on other optical imaging systems (for example, single plane illumination microscopes[41], fluorescence endoscopes[42]).

Compared to alternative methods implemented here for validation purposes, the fluorescent actinometers benefit from (1) direct access to information sought for, independently on their concentration and without any further measurement (for example, the illuminated surface); (2) measuring light intensities at the surface of samples as well as in their depth, which is difficult to obtain by any other method. Hence irradiance could be calibrated with depth in thick samples by using gels with similar refractive index as biological tissues; (3) a high signal-to-noise ratio from using fluorescence; (4) online access to actinometer properties, (https://chart-studio.plotly.com/~Alienor134/#/) and codes and user-friendly applications for data processing (https://github.com/DreamRepo/light_calibration) and (5) easy and fast (less than 1 hour, from sample preparation to data processing) transfer of know-how to end users. Their kinetics-based protocol is notably robust with respect to parameters that may affect fluorescence from the sample[43] and the instrument[44] side[13]. In particular, the quantum yield of fluorescence of the actinometer does not enter into the expression of the characteristic time τ. Hence, its temperature and possible wavelength dependence are not detrimental as long as the temperature and wavelength of light excitation remain constant over the measurement. With 20% measurement uncertainty, even a spatial gradient of 20 °C over the distance overcome in time τ by the reporting fluorophore, for example rhodamine B[45], would not affect the result.

However, the time resolution of the fluorescent actinometers is fixed by the kinetics of their photoconversion, which imposes a lower limit on both the measurement duration (still short, 300 ms–30 min) and the ability to analyze time evolving profiles of light intensity. Moreover, the mechanisms underlying their photoconversion often involve multiple steps. Thus, the mechanistic reduction making relevant monoexponentially fitting the time evolution of their fluorescence response to illumination[16] is reliable only in ranges of light intensity in which the light-driven photoconversion step is constant and rate limiting.

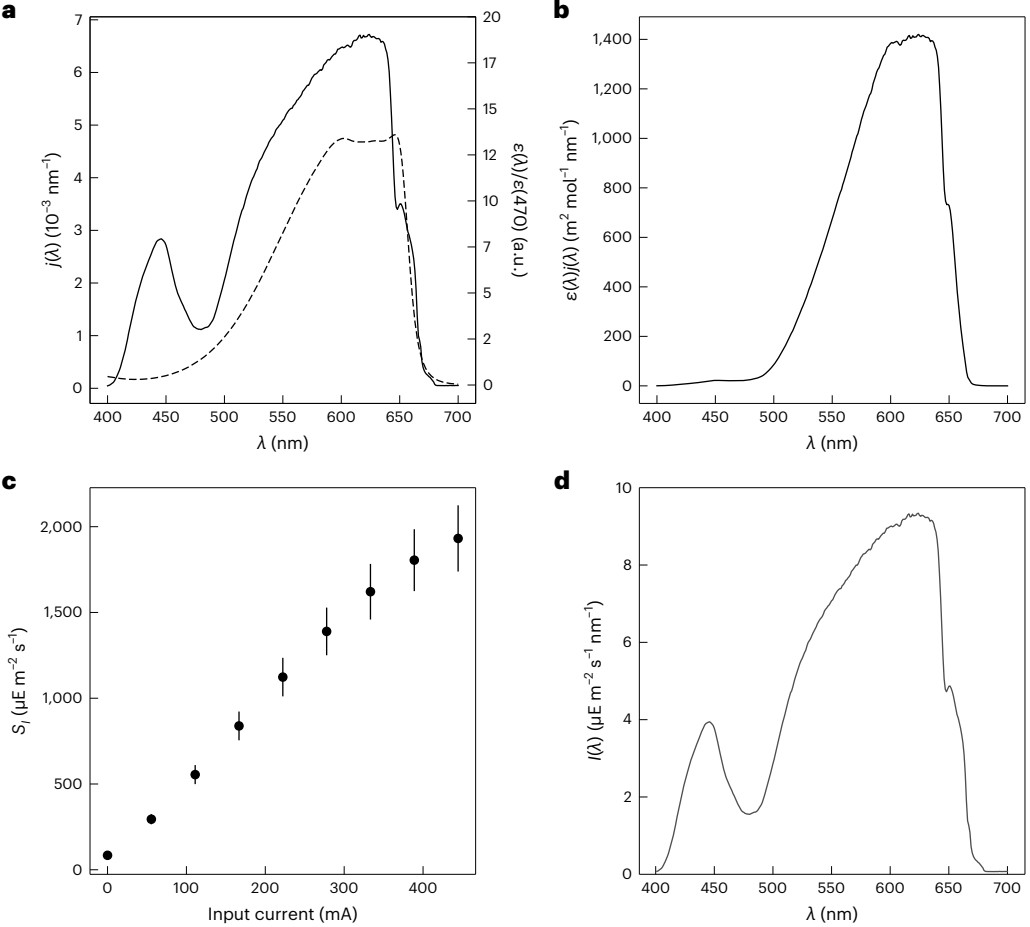

**Fig. 5 | Characterization of the spectral light intensity of a white LED by DDAO-mediated measurement. a**, Emission spectrum of the white LED normalized by its integral (solid line), absorbance spectrum of DDAO normalized at 470 nm (dashed line). **b**, Action spectrum of the white LED on DDAO. **c**, Dependence of the integral light intensity $S_I$ emitted by the white LED on the input currents as measured with spectral transfer of DDAO fluorescence intensity (mean value ± s.d., s.d. are propagated from the error on $\sigma(470)$ (20 repeats)). **d**, Incident spectral light intensity $I(\lambda)$ of the white LED fed with 277 mA current as retrieved with DDAO ($S_I = 1.4$ mE m$^{-2}$ s$^{-1}$) (text and Supplementary Table 1).

Eventually, two issues arise for characterizing spatially inhomogeneous light profiles: (1) as illustrated by imaging a Petri dish containing a Dronpa-2 solution illuminated by a surrounding radial array of LEDs (Supplementary Note 9), a 3D-inhomogeneous light profile cannot be retrieved from the 2D-map of the characteristic time $\tau$. Yet, one can still retrieve a useful information on the light intensity averaged over the sample thickness (Extended Data Figs 6 and 7); (2) as shown in Fig. 2g–i, molecular motion introduces blurring in the retrieved 2D light profile. However, this phenomenon is marginal as long as molecular diffusion is spatially limited at the $\tau$ time scale, which can be reached by using a medium in which molecular diffusion is reduced. A polyacrylamide gel is relevant for the water-soluble fluorescent actinometers. Immobilization of the synthetic fluorescent actinometers would require developments, which have not been covered during the present study.

We envision that such organic dye- and fluorescent protein-based actinometry will improve our understanding of how light dose effects the health and viability of biological specimens, and think that measuring and reporting light dose experienced by the sample should become commonplace for improving reporting and reproducibility in microscopy.

## Online content

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

## Methods

The Supplementary Information starts with a list, which indicates the (sub)sections associated with the supplementary elements of the main text, as well as details on statistical parameters and image processing. It is then divided in two parts: (1) sections 1–3 are dedicated to end users, who want to directly implement the reported fluorescent actinometers; and (2) sections 4–9 contain more advanced information as well as the complete validation of the reported fluorescent actinometers. Codes and metadata for the figures in the main text can be found online, as well as user-friendly tools and how to use them: https://github.com/Dream-Repo/light_calibration and in a mirror Zenodo repository (https://doi.org/10.5281/zenodo.7966573).

### Syntheses

**Cin.** Bromine (2.32 g, 0.74 ml, 14.5 mmol; 2 eq.) was added dropwise to a solution of 2,4-dihydroxybenzaldehyde (1.00 g, 7.25 mmol) in acetic acid (10 ml) over 30 min at room temperature. The resulting mixture was vigorously stirred for 2 h at room temperature. After addition of water (20 ml), the precipitate was filtered, washed with water and dried. 3,5-dibromo-2,4-dihydroxybenzaldehyde was obtained as pale orange crystals after recrystallization of the crude residue in ethanol (1.20 g, 55% yield). m.p: 200 °C; $^1$H-nuclear magnetic resonance (NMR) (ppm, 250 MHz, CDCl$_3$, 298 K) δ 9.68 (s, 1 H), 7.70 (s, 1 H), 6.60 (s, 1 H); $^{13}$C-NMR (ppm, 62.8 MHz, CDCl$_3$, 298 K) δ 193.5, 158.6, 157.1, 135.6, 114.9, 99.9 and 97.8.

Ethyl bromoacetate (5.75 g, 34.5 mmol) was added to a solution of triphenylphosphine (10.0 g, 38 mmol; 1.15 eq) in toluene (40 ml). The mixture was vigorously refluxed for 10 h under stirring. The white precipitate was filtered off, washed with toluene and dried. The addition at 5 °C of 1 M NaOH (50 ml) to a solution of white solid (10 g) in water (200 ml) gave a white and gummy solid that was filtered, washed with water and dried to yield 1-carboxymethylidene triphenyl phosphorane as a white solid (8.0 g, 60% yield).

A mixture of 3,5-dibromo-2,4-dihydroxybenzaldehyde and 1-carboethoxymethylidene triphenyl phosphorane (1.5 eq) in toluene (10 ml for 1 mmol of aldehyde) was heated at 60 °C under argon on protecting from light. The course of the reaction was followed by cyclohexane and AcOEt. After 2 to 4 h, the reaction was completed. After cooling to room temperature, toluene was removed in a vacuum. The crude residue was purified by flash chromatography on silica gel (mixtures of ethyl acetate and cyclohexane as eluent) to give Cin in 40 to 60% yield[15]. m.p. 118–118.5 °C; $^1$H-NMR (ppm, 250 MHz, CDCl$_3$, 298 K) δ 7.81 (d, 1 H, $J$ = 16.1 Hz), 7.60 (s, 1 H), 6.47 (d, 1 H, $J$ = 16.1 Hz), 6.07 (bs, 2 H), 4.22 (q, 2 H, $J$ = 7.0 Hz), 1.33 (t, 3 H, $J$ = 7.0 Hz); $^{13}$C-NMR (ppm, 62.8 MHz, CD$_3$COCD$_3$, 298 K) δ 167.3, 154.2, 153.7, 138.7, 131.6, 118.8, 117.8, 102.1, 101.8, 60.7, 14.6. Elemental analysis: (%) for C$_{11}$H$_{10}$O$_4$Br$_2$(365.9): C 36.10, H 2.75; found: C 36.06, H 2.63; mass spectrometry: MS (CI, CH$_4$): $m/z$ 367 [M + 1]; MS (CI, CH$_4$, HR): $m/z$ 364.9024, 366.9006 and 368.8992 (calculated mass for C$_{11}$H$_{10}$O$_4$Br$_2$: 364.9024, 366.9004 and 368.8985).

**Nit.** 4-(Diethylamino)benzaldehyde (1.5 g, 8.5 mmol) and phenylhydroxylamine (0.925 g, 8.5 mmol) were stirred in glacial acetic acid (8 ml) at room temperature for 2 h. The reaction mixture was then poured into water and extracted with ether. The ether extracts washed with saturated aqueous sodium bicarbonate and with brine, dried on sodium sulfate and concentrated under a vacuum. After recrystallization in cyclohexane and toluene, Nit[24] was obtained as orange needles (1.70 g, 6.4 mmol, 75%). $^1$H-NMR (CDCl$_3$, 300 MHz) δ 8.30 (d, 2H, $J$ = 9 Hz); 7.79–7.76 (m, 3H), 7.48–7.37 (m, 3H), 6.71 (d, 2H, $J$ = 9 Hz), 3.43 (q, 4H, $J$ = 7 Hz), 1.22 (t, 6H, $J$ = 7 Hz).

**DASA.** 3-Methyl-1-(4-sulfophenyl)-5-pyrazolone (1.97 mmol, 500 mg) was suspended in water (5 ml) and carefully neutralized to pH 7 with 1 M sodium bicarbonate. To this, furfural (1.97 mmol, 189 mg) was added and the reaction mixture stirred at 20 °C for 16 h. The solvent was then evaporated under reduced pressure to give the sodium 4-(4-(furan-2-ylmethylene)-3-methyl-5-oxo-4,5-dihydro-1H-pyrazol-1-yl)benzenesulfonate intermediate (I) as a red solid (494 mg, 76% yield). The product is a mixture of Z and E isomers (2:1). $^1$H-NMR (300 MHz, deuterium oxide) δ 8.09 (d, $J$ = 3.8 Hz, 1H), 7.83 (d, $J$ = 1.6 Hz, 1H), 7.80–7.61 (m, 7H), 7.33 (s, 1H), 7.18 (d, $J$ = 3.5 Hz, 0.5H), 7.16 (s, 0.5H), 6.68 (dd, $J$ = 3.7, 1.6 Hz, 1H), 6.61 (dd, $J$ = 3.6, 1.7 Hz, 0.5H), 2.32 (s, 1.5H), 2.11 (s, 3H). $^{13}$C-NMR (75 MHz, DMSO) δ 164.32, 161.54, 150.90, 150.76, 150.26, 150.11, 149.51, 148.12, 144.35, 138.07, 130.29, 127.36, 127.22, 126.24, 126.18, 124.65, 121.29, 120.10, 116.91, 116.62, 114.84, 114.45, 17.39, 12.64. High-resolution mass spectrometry (electrospray ionization+) $m/z$ calculated. For [C$_{15}$H$_{11}$N$_2$O$_5$S] [M−H+]: 331.039; found, 331.040.

Compound I (0.565 mmol, 200 mg) and indoline (0.565 mmol, 67 mg) were dissolved in methanol (2 ml) and stirred at 20 °C for 1 h. Then this solution was diluted in 10 ml of ethyl ether, the precipitate filtered and washed with 3 × 5 ml of ethyl ether to yield DASA[17] as a dark blue powder (165 mg, 62% yield). $^1$H-NMR (300 MHz, deuterium oxide) δ 8.01 (dd, $J$ = 6.0, 1.9 Hz, 1H), 7.95 (d, $J$ = 8.4 Hz, 2H), 7.76 (d, $J$ = 8.6 Hz, 2H), 7.27 (d, $J$ = 7.6 Hz, 1H), 7.05 (t, $J$ = 7.6 Hz, 1H), 6.86 (t, $J$ = 7.3 Hz, 1H), 6.67–6.56 (m, 2H), 5.37 (d, $J$ = 1.5 Hz, 1H), 3.76 (d, $J$ = 3.2 Hz, 1H), 3.48–3.37 (m, 2H), 3.15–2.97 (m, 2H), 1.80 (s, 3H). $^{13}$C-NMR (75 MHz, DMSO) δ 202.51, 163.24, 162.70, 150.52, 149.47, 144.27, 137.06, 134.63, 129.83, 126.99, 126.28, 124.53, 117.63, 116.97, 107.38, 103.62, 61.04, 47.83, 43.50, 27.78 and 10.58. Additional small peaks due to the presence of the keto isomer on the pyrazole. High-resolution mass spectrometry (electrospray ionization−) $m/z$ calculated. For [C$_{23}$H$_{20}$N$_3$O$_5$S] [M−]: 450.11; found, 450.11.

### Production of Dronpa-2-containing samples

**Plasmids.** The plasmids for bacterial expression of Dronpa-2 carrying an N-terminal hexahistidine tag and for mammalian expression of Dronpa-2 fused at the C terminal of the histone H2B (H2B-Dronpa-2) have been previously described in refs. 39,41.

**Production of Dronpa-2-labeled *E. coli*.** *E. coli* cells from the TOP10 strain were transformed with the Dronpa-2 plasmid by electroporation. The transformed *E. coli* cells were grown at 37 °C in Luria Bertani medium. When the optical density at 600 nm reached 0.2, expression was induced by addition of isopropyl β-D-1-thio-galactopyranoside (IPTG) to a final concentration of 1 mM. After 4 h of expression at 30 °C, 1 ml aliquots were taken and cells were centrifuged at 8,000 rpm for 5 min. After centrifugation, the supernatant was removed and the *E. coli* cells were washed once with 1 ml of PBS (pH 7.4, 50 mM sodium phosphate, 150 mM NaCl) and then resuspended in 250 µl of PBS buffer.

This suspension of *E. coli* was used to prepare a monolayer of bacterial cells deposited on an agarose pad as follows: 125 µl of a 2% pad of low-melting agarose in PBS was sandwiched between two circular glass slides separated by 250 µm by spacers (Gene Frames AB0578; Thermo Scientific). After the agarose became solid, the top cover slide was removed and 2 µl of the bacterial suspension was deposited on the surface of the agar pad. After 15 min, the top cover slide was replaced to seal the sample.

**Production and purification of Dronpa-2.** The Dronpa-2 plasmid with an N-terminal hexahistidine tag was transformed in *E. coli* BL21 strain. Cells were grown in Terrific Broth at 37 °C. The expression was induced at 30 °C or 16 °C by addition of IPTG to a final concentration of 1 mM at optical density at 600 nm of 0.6. The cells were collected after 16 h of expression and lysed by sonication in lysis buffer (50 mM PBS with 150 mM NaCl at pH 7.4, 1 mg ml$^{-1}$ DNAse, 5 mM MgCl$_2$ and 1 mM phenylmethylsulfonyl fluoride, and a cocktail of protease inhibitors (Sigma Aldrich; catalog no. S8830)). After lysis, the mixture was incubated on ice for 2 h for DNA digestion. The insoluble material was removed by centrifugation and the supernatant was incubated overnight with Ni-NTA agarose beads (Thermo Fisher) at 4 °C in a rotator-mixer. The

protein loaded Ni-NTA column was washed with 20 column volumes of N1 buffer (50 mM PBS, 150 mM NaCl, 20 mM imidazole, pH 7.4) and 5 column volumes of N2 buffer (50 mM PBS, 150 mM NaCl, 40 mM imidazole, pH 7.4). The bound protein was eluted with N3 buffer (50 mM PBS, 150 mM NaCl, 0,5 M imidazole, pH 7.4). The protein fractions were eventually dialyzed with cassette Slide-A-Lyzer Dialysis Cassettes (Thermo Fisher) against 50 mM PBS, 150 mM NaCl pH 7.4.

**Production of Dronpa-2-labeled mammalian cells.** U-2 OS cells were grown at 37 °C in 5% $CO_2$ in air atmosphere in McCoy's 5A Medium complemented with 10% fetal bovine serum. Cells were transiently transfected with Genejuice (Merck) according to the manufacturer's protocol then washed with Dulbecco's phosphate buffered saline (2.7 mM KCl, 138 mM NaCl, 1.5 mM $KH_2PO_4$, 8.1 mM $Na_2HPO_4$, Thermo Fisher) and fixed with 2% paraformaldehyde solution in Dulbecco's phosphate buffered saline.

**Production of photosynthetic apparatus-containing samples**
The algae strain used were wild type CC124 and WT4 of *Chlamydomonas reinhardtii* provided by the Institut de Biologie Physico-Chimique (http://www.ibpc.fr/UMR7141/en/home/). The algae were grown in heterotrophic media TAP (https://www.chlamycollection.org/methods/media-recipes/tap-and-tris-minimal/) under constant 5–10 μE m$^{-2}$ s$^{-1}$ white LED and agitation at 25 °C. The population was diluted to one-tenth the day before the experiment to ensure that the culture observed is in exponential phase.

### Reporting summary
Further information on research design is available in the Nature Portfolio Reporting Summary linked to this article.

## Data availability
The online repository contains representative raw data files corresponding to the methods described. The datasets generated and/or analyzed during the current study and that are not in the online repository due to their profuse nature are available from the corresponding author on request. Absorption and emission spectra of the actinometers: https://github.com/DreamRepo/light_calibration/tree/main/spectra_plotly and https://chart-studio.plotly.com/~Alienor134/#/. Metadata for the video acquisitions used to produce the main text figures: https://github.com/DreamRepo/light_calibration/tree/main/imaging_metadata. Simulations of the illumination used in Fig. 2k,l: https://github.com/DreamRepo/light_calibration/tree/main/Macroscope. Simulation of 3D illumination pattern and comparison with 2D imaging: https://github.com/DreamRepo/light_calibration/tree/main/LED%20Array. Representative data: https://github.com/DreamRepo/light_calibration/blob/main/data.

## Code availability
The codes that require several steps to reproduce the data analysis have been made available online: https://github.com/DreamRepo/light_calibration and in a mirror Zenodo repository (https://doi.org/10.281/zenodo.7665939). Application to process a fluorescence evolution curve: https://github.com/DreamRepo/light_calibration/releases/tag/review. Wavelength transfer using DDAO: https://github.com/DreamRepo/light_calibration/tree/main/DDAO. Light intensity in the field of view (Fig. 2): jupyter notebook: https://github.com/DreamRepo/light_calibration/blob/main/notebooks/Dronpa2_video.ipynb, instructions: https://github.com/DreamRepo/light_calibration/tree/main. Implementation of the fitting algorithm for the fluorescence induction of microalgae: https://github.com/DreamRepo/light_calibration

/blob/main/notebooks/PA_OJIP_rise_fit.ipynb, data: https://github.com/DreamRepo/light_calibration/blob/main/data/2022-01-17_13_33_ojip_curve_363_blue.csv. The python codes can be executed without preliminary installation with Binder: https://mybinder.org/v2/gh/DreamRepo/light_calibration/HEAD.

## Acknowledgements
We thank N. Gagey for providing Cin, S. Bujaldon for providing *Chlamydomonas reinhardtii* strains, S. Lahlou for the seagull picture used for patterning the light and A. Jawahar for useful discussions on the protocol implementation. D. Bensimon, C. Boccara, P. Dedecker, G. Ellis-Davies and L. Fensterbank are acknowledged for manuscript reading and discussions. This work was supported by the ANR (grant nos. France BioImaging, ANR-10-INBS-04; Morphoscope2, ANR-11-EQPX-0029 and IPGG, ANR-10-IDEX-0001-02 PSL, ANR-10-LABX-31 and ANR-19-CE11-0005 for T.L.S. and L.J.), the Federal Ministry of Education and Research of Germany (BMBF) within the YESPVNIGBEN project (grant no. 03SF0576A for Y.N.), the European Regional Development Fund project 'Plants as a tool for sustainable global development' (grant no. CZ.02.1.01/0.0/0.0/16_019/000082 7 for D.L. and L.N.) and the European Innovation Council Pathfinder Open DREAM (grant no. 101046451 for A.L., I.C., T.L.S., L.J., L.N., D.L. and V.C.).

## Author contributions
Conceptualization was done by A.L., T.L.S. and L.J. Methodology was done by A.L., L.N., T.L.S. and L.J. Software was provided by A.L., I.C. and T.L.S. Validation was carried out by A.L., H.S.T., I.C. and D.L. Formal analysis was done by A.L., H.S.T., I.C., T.L.S. and L.J. Investigation was carried out by A.L., H.S.T., I.C., M.M., Y.N., I.E., R.J., P.M. and T.L.S. Resources were provided by A.L., I.C., Y.S., M.-A.P., I.A., Y.N., L.N., P.M., W.S., E.B., I.E., N.D., V.C. and R.J. Data were curated by A.L., I.C., Y.N., P.M. and T.L.S. The original draft was written by A.L., H.S.T., I.C., L.N., D.L., T.L.S. and L.J. Review and editing of the draft were done by A.L., T.L.S. and L.J. Visualization was done by A.L., H.S.T., T.L.S. and L.J. Supervising the project were A.L., T.L.S. and L.J. Project administration was done by L.J. Funding was acquired by T.L.S. and L.J.

## Competing interests
The authors declare no competing interests.

## Additional information
**Extended data** is available for this paper at https://doi.org/10.1038/s41592-023-02063-y.

**Correspondence and requests for materials** should be addressed to Aliénor Lahlou, Thomas Le Saux or Ludovic Jullien.

**a**

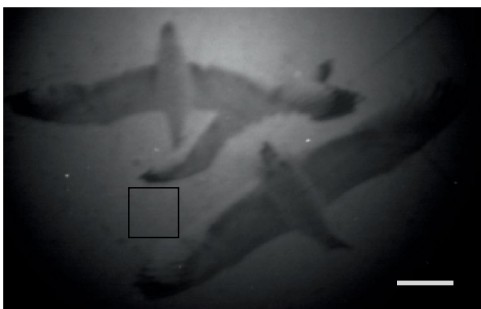

**b**

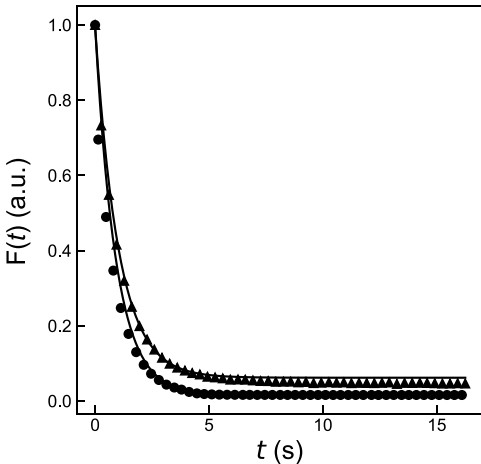

**Extended Data Fig. 1 | Comparison of the fluorescence decay associated to the photoconversion of Dronpa-2 in solution and in a polyacrylamide gel. a**: 100×100 pixels region of interest selected for analysis in the image generated in the polyacrylamide gel (scale bar: 100 μm); **b**: Averaged decays of the fluorescence signal over the selected area. Experimental data: disk (solution) triangles (gel); monoexponential fits: solid lines (solution: τ = 0.91 s, polyacrylamide gel: τ = 1.03 s).

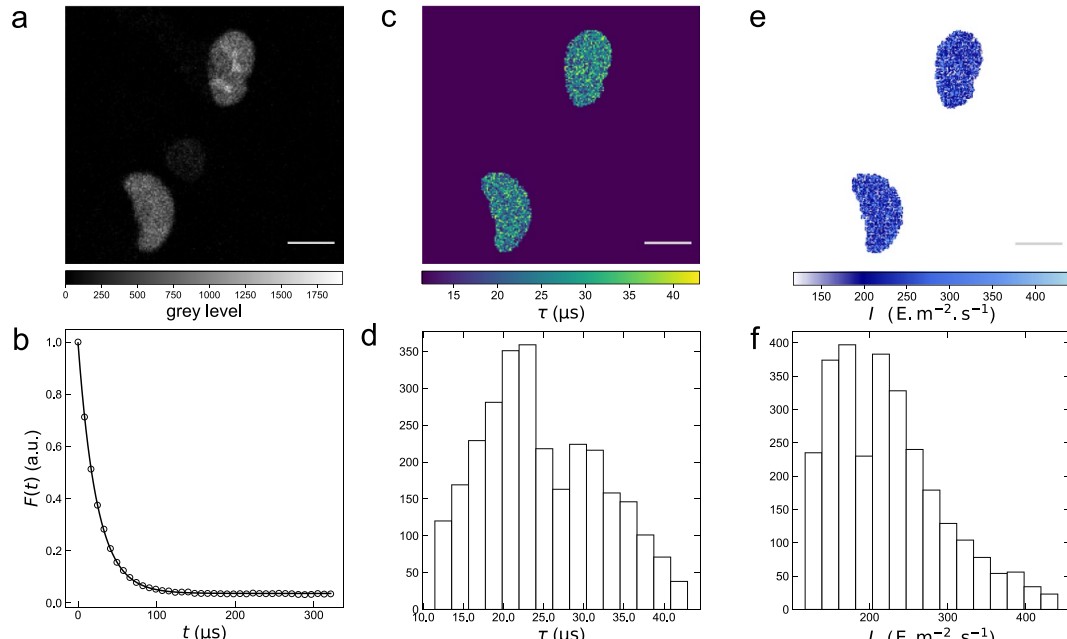

**Extended Data Fig. 2 | Dronpa-2 for characterization of the spatial distribution of the light intensity of a confocal microscope equipped with a continuous laser. a, b**: Dronpa-2-labeled nucleus of a fixed U-2 OS cell imaged with a confocal microscope in the raster scanning mode ($\lambda_{exc}$ = 488 nm; 500 nm ≤ $\lambda_{em}$ ≤ 600 nm). Initial image (**a**); Time evolution of the averaged fluorescence over the whole nucleus (**b**; circles: experimental data; solid line: monoexponential fit with Eq.(S1). The corresponding evolution from a central portion of the overall image of a 10 µM Dronpa-2 solution sandwiched between two glass-slides is shown with circles); **c–f**: Maps of the characteristic time τ (**c**) and light intensity (**d**), and corresponding histograms (**e,f**; a 3×3 binning is applied to the initial video sequence to improve fitting accuracy). Solvent: Tris buffer pH 7.4 (50 mM Tris, 150 mM NaCl); Pixel size: 0.33 µm; Laser power 10%; $T$ = 293 K ; Scale bar: 29 µm; Independent repeats: 4.

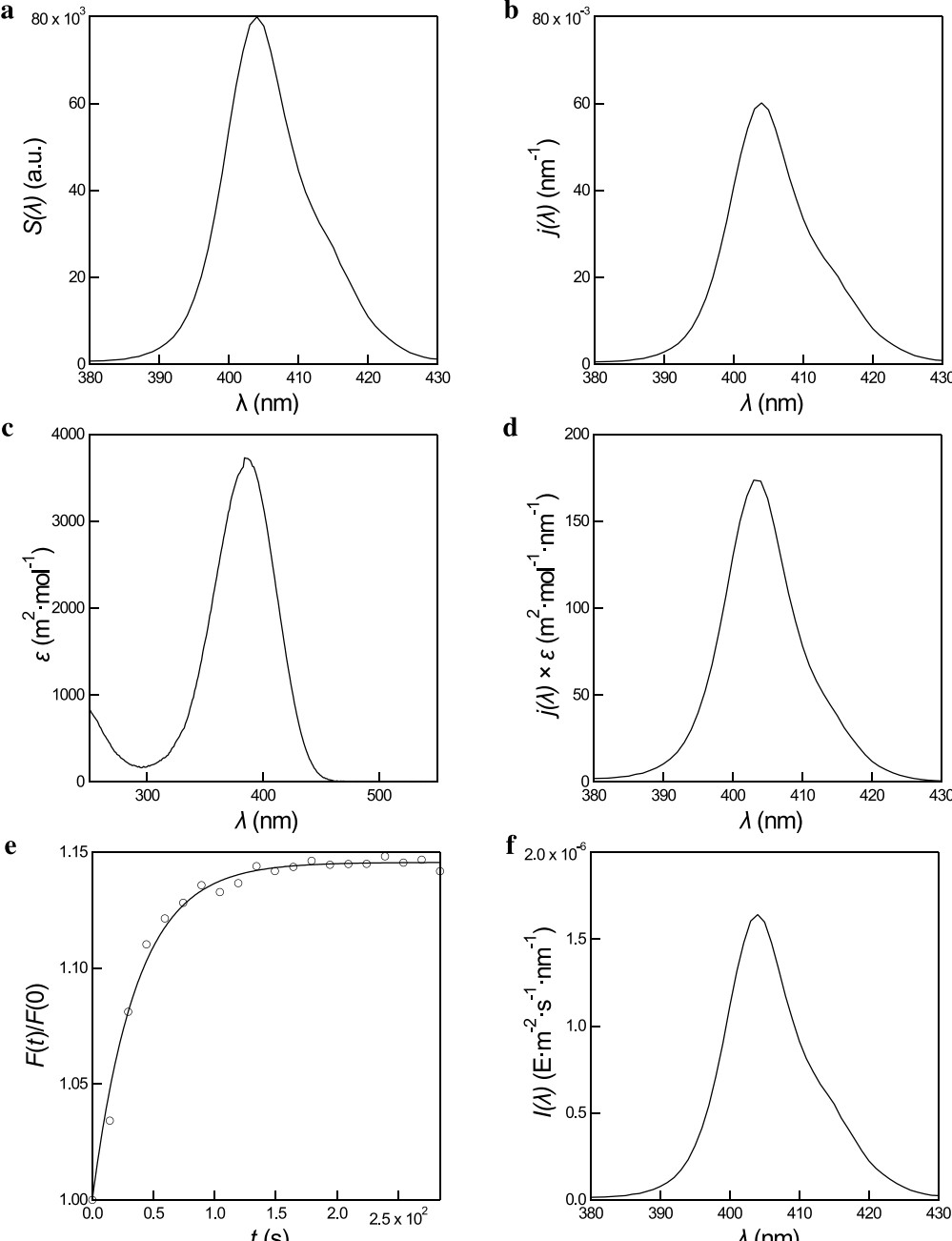

**Extended Data Fig. 3 | Quantitation of a purple LED emitting at 405 nm.**
Non-normalized ($S(\lambda)$; **a**) and normalized ($j(\lambda) = S(\lambda)/S$; **b**) emission spectrum of the LED at 405 nm; **c**: Absorption spectrum of **Nit** $\varepsilon(\lambda)$; **d**: Action spectrum of the LED on **Nit** in ethanol; **e**: Rise of the fluorescence emission of a 12 μM **Nit** and 1 μM RhB solution in ethanol at 574 nm as a function of time. Markers: experimental data; solid line: monoexponential fit. From the fit, we retrieved $\tau = 38.3$ s for the relaxation time. T = 293 K; **f**: Scaled emission spectrum of the LED.

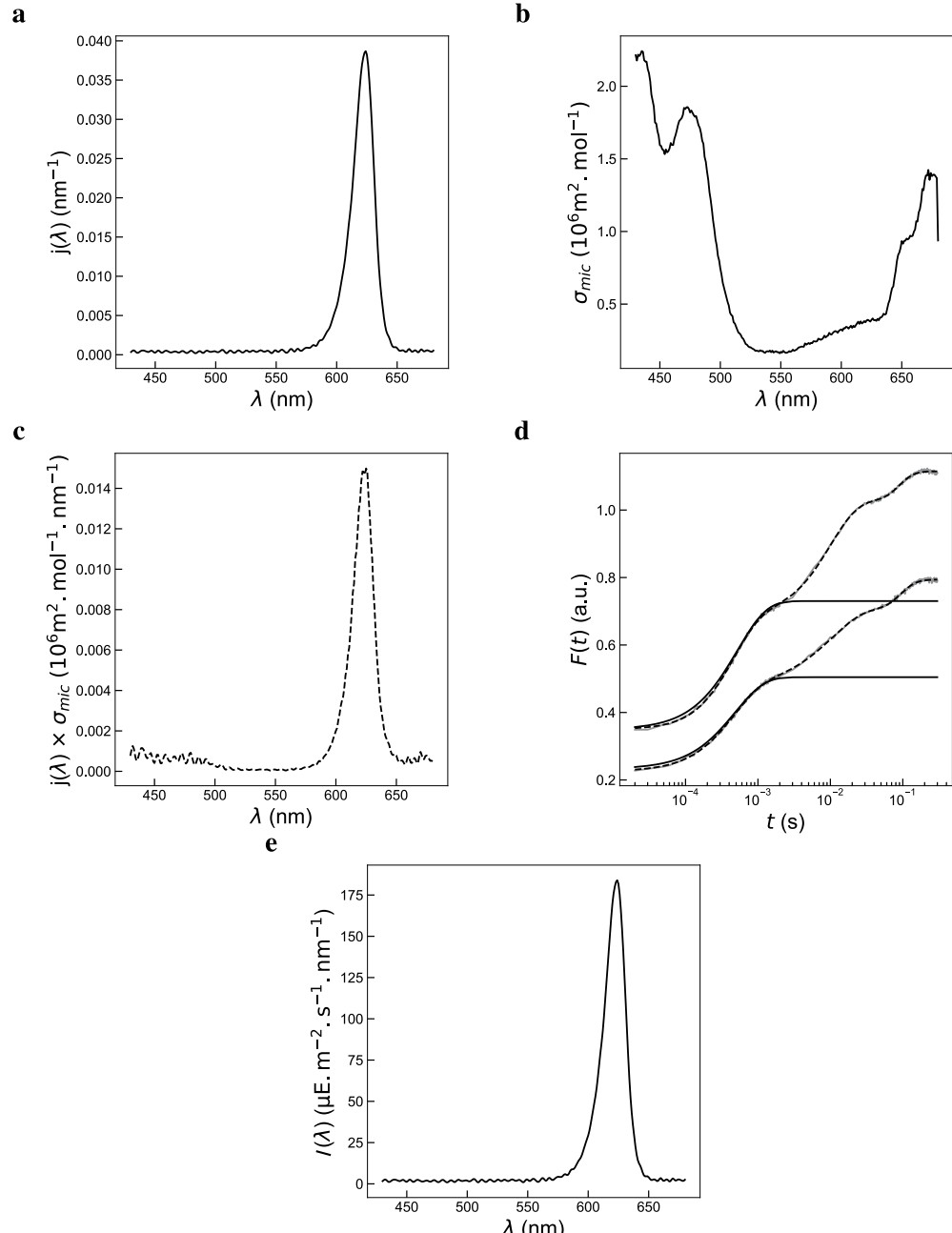

**Extended Data Fig. 4 | Quantitation of a red-orange LED emitting at 625 nm.**
**a**: Normalized emission spectrum ($j(\lambda) = S(\lambda)/S$) of the LED at 625 nm; **b**: Scaled fluorescence excitation spectrum $\sigma_{mic}(\lambda)$ of PA in *Chlamydomonas reinhardtii* (CC 124) in exponential phase in minimal media ($\lambda_{em} = 470$ nm); **c**: Action spectrum of the LED at 625 nm on PA in *Chlamydomonas reinhardtii* (CC 124); **d**: Time evolution of the PA fluorescence emission under constant illumination at 625 nm

for the power setting 60% on the PSI instrument (2 repeats). The experimental data (grey markers) have been fitted with Eqs.(S3) (dashed lines) and (S4) (solid lines) to retrieve $\tau = 427$ µs and $\tau = 460$ µs for the characteristic time $\tau$ associated to the initial step of PA fluorescence rise ; **e**: Scaled emission spectrum of the LED at 625 nm. $T = 293$ K.

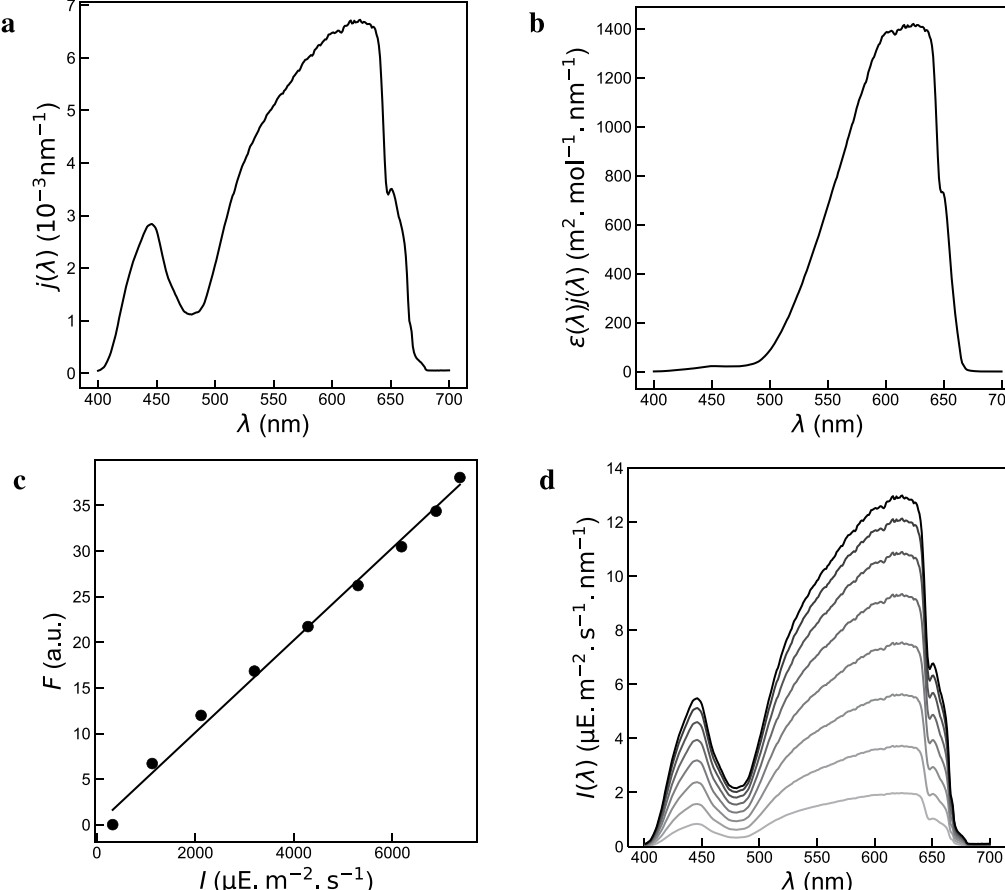

**Extended Data Fig. 5 | Quantitation of a white LED with DDAO. a**: Emission spectrum of the white LED normalized by its integral $j(\lambda) = S(\lambda)/S$; **b**: Action spectrum of the white LED on DDAO $\epsilon(\lambda)j(\lambda)$; **c**: Dependence of the fluorescence level of 10 µM DDAO in aqueous HEPES pH 7.9 buffer (100 mM NaCl, 5 mM NaOH, 10 mM HEPES) sandwiched between two microscope slides F ($\lambda_{exc,1}$) on the light intensity I ($\lambda_{exc,1}$) at 470 nm. Squares: experimental data from averaging the fluorescence collected by the camera, solid line: linear fit; **d**: Scaled spectral photon flux of the white LED for current levels feeding the white LED ranging from 55, 111, 166, 222, 277, 333, 388, 444 mA (light to dark). $T$ = 293 K.

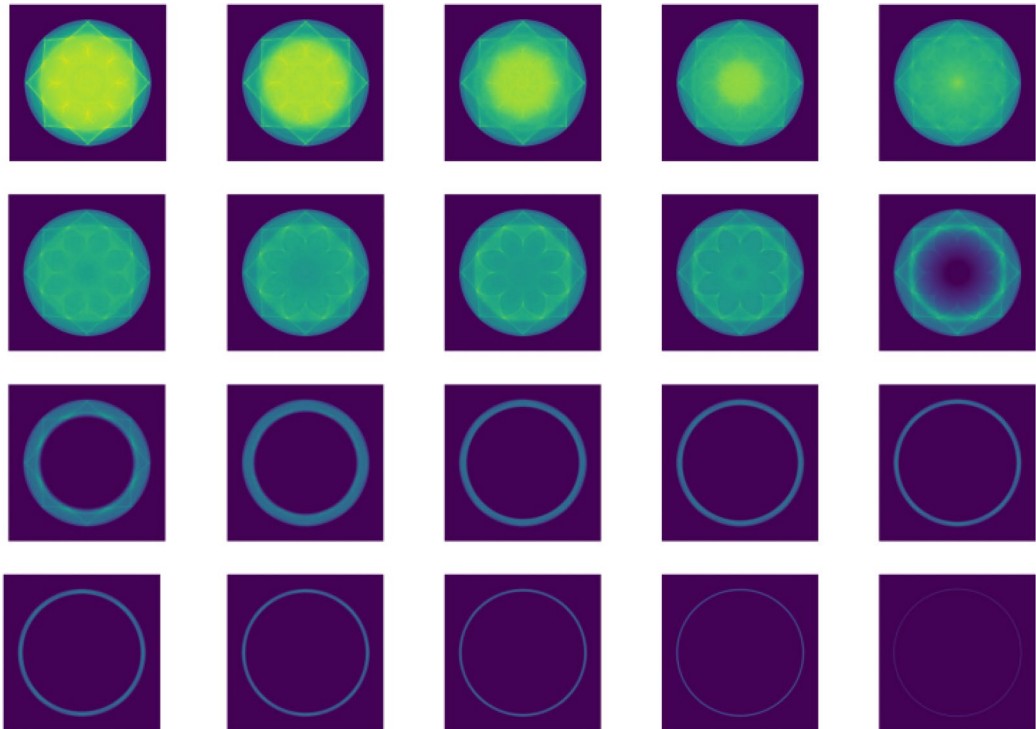

**Extended Data Fig. 6 | 2D images of the absorbed light flux at each of the 20 z-axis locations within the detection volume.** The first image corresponds to the **Dronpa-2** solution space adjacent to the bottom of the Petri dish, and the following images, going left to right, and down, in a raster fashion, correspond to image slices progressively closer to the fluid surface. Since this work did not consider the absolute light intensity, but rather the distribution, a color scale is omitted.

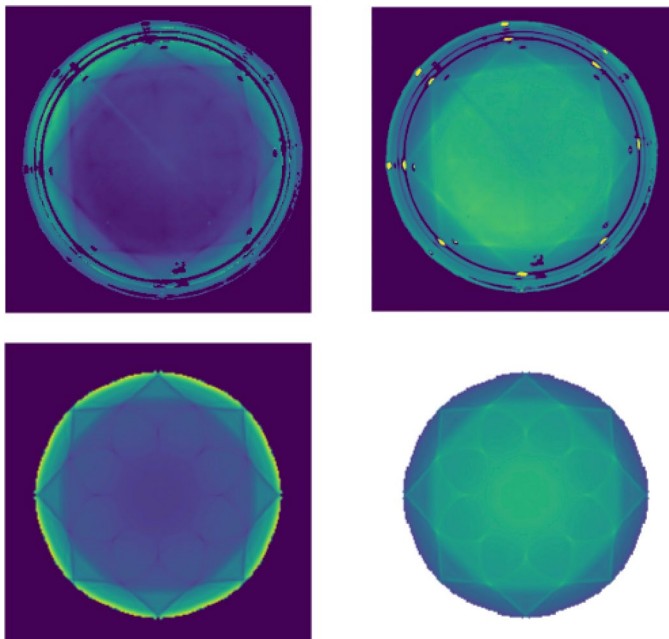

**Extended Data Fig. 7 | Experimental and simulated intensity maps in a 3D sample.** τ (left) and light intensity (right) maps of the experimental (top) and simulation (bottom) situation. Since this work did not consider the absolute light intensity, but rather the distribution, a color scale is omitted.

# Reporting Summary

## Statistics

For all statistical analyses, confirm that the following items are present in the figure legend, table legend, main text, or Methods section.

| n/a | Confirmed | |
|---|---|---|
| ☐ | ☒ | The exact sample size (*n*) for each experimental group/condition, given as a discrete number and unit of measurement |
| ☒ | ☐ | A statement on whether measurements were taken from distinct samples or whether the same sample was measured repeatedly |
| ☒ | ☐ | The statistical test(s) used AND whether they are one- or two-sided<br>*Only common tests should be described solely by name; describe more complex techniques in the Methods section.* |
| ☒ | ☐ | A description of all covariates tested |
| ☒ | ☐ | A description of any assumptions or corrections, such as tests of normality and adjustment for multiple comparisons |
| ☐ | ☒ | A full description of the statistical parameters including central tendency (e.g. means) or other basic estimates (e.g. regression coefficient) AND variation (e.g. standard deviation) or associated estimates of uncertainty (e.g. confidence intervals) |
| ☒ | ☐ | For null hypothesis testing, the test statistic (e.g. *F*, *t*, *r*) with confidence intervals, effect sizes, degrees of freedom and *P* value noted<br>*Give P values as exact values whenever suitable.* |
| ☒ | ☐ | For Bayesian analysis, information on the choice of priors and Markov chain Monte Carlo settings |
| ☒ | ☐ | For hierarchical and complex designs, identification of the appropriate level for tests and full reporting of outcomes |
| ☒ | ☐ | Estimates of effect sizes (e.g. Cohen's *d*, Pearson's *r*), indicating how they were calculated |

*Our web collection on statistics for biologists contains articles on many of the points above.*

## Software and code

Policy information about availability of computer code

| Data collection | UV/Vis spectrophotometer: Cary 300 UV-Vis SOFTWARE Cary Win UV VERSION 4.10(464)<br>Fluorescence spectrophotometer: LPS 220 spectrofluorometer SOFTWARE PTI Felix VERSION 4.1.0.3096<br>Custom epifluorescence microscope/macroscope: Python 3.9.10 - Conda 22.9.0. The code will be published in 2023/2024 in a hardware-related publication.<br>Pulsed confocal microscope: Leica TCS SP SOFTWARE LAS X<br>Continuous confocal microscope: Zeiss LSM 710 Laser Scanning Microscope SOFTWARE ZEN 2012 SP5 FP3 VERSION 14.0.20.201<br>One photon excitation FCS: Custom built set-up Igor Pro 8 or 9 |
|---|---|
| Data analysis | NMR: Mnova Mestrelab 8.1.2-11880<br>Spectrophotometer/fluorometer: Igor v8-v9 and python 3.9.10<br>FCS: Igor Pro 8 or 9<br>RICS: Global software for Images SimFCS 4, MetroloJ (last online modification: 06/07/17)<br>Microscopy images: Python 3.9.10, Conda 22.9.0, package versions specified in https://github.com/DreamRepo/light_calibration/blob/main/binder/requirements.txt<br>Optical simulations: Zemax - Optic Studio 18.9 |

For manuscripts utilizing custom algorithms or software that are central to the research but not yet described in published literature, software must be made available to editors and reviewers. We strongly encourage code deposition in a community repository (e.g. GitHub). See the Nature Portfolio guidelines for submitting code & software for further information.

## Data

Policy information about availability of data

All manuscripts must include a data availability statement. This statement should provide the following information, where applicable:
- Accession codes, unique identifiers, or web links for publicly available datasets
- A description of any restrictions on data availability
- For clinical datasets or third party data, please ensure that the statement adheres to our policy

The online repository (https://github.com/DreamRepo/light_calibration and https://doi.org/10.5281/zenodo.7966573) contains representative raw data files corresponding to the methods described. The datasets generated and/or analyzed during the current study and that are not in the online repository due to their profuse nature are available from the corresponding author on request.

Absorption and emission spectra of the actinometers https://github.com/DreamRepo/light_calibration/tree/main/spectra_plotly and https://chart-studio.plotly.com/~Alienor134/#/
Metadata for the video acquisitions used to produce the main text figures: https://github.com/DreamRepo/light_calibration/tree/main/imaging_metadata
Simulations of the illumination used in Figure 2k,l: https://github.com/DreamRepo/light_calibration/tree/main/Macroscope
Simulation of 3D illumination pattern and comparison with 2D imaging: https://github.com/DreamRepo/light_calibration/tree/main/LED%20Array
Application to process a fluorescence evolution curve: https://github.com/DreamRepo/light_calibration/releases/tag/review with example data: https://github.com/DreamRepo/light_calibration/blob/main/data/video_light.tiff
Wavelength transfer using DDAO: https://github.com/DreamRepo/light_calibration/tree/main/DDAO
Light intensity in the field of view (Figure 2): jupyter notebook: https://github.com/DreamRepo/light_calibration/blob/main/notebooks/Dronpa2_video.ipynb, instructions: https://github.com/DreamRepo/light_calibration/tree/main
Implementation of the fitting algorithm for the fluorescence induction of microalgae: https://github.com/DreamRepo/light_calibration/blob/main/notebooks/PA_OJIP_rise_fit.ipynb data: https://github.com/DreamRepo/light_calibration/blob/main/data/2022-01-17_13_33_ojip_curve_363_blue.csv

## Human research participants

Policy information about studies involving human research participants and Sex and Gender in Research.

| | |
|---|---|
| Reporting on sex and gender | N/A |
| Population characteristics | N/A |
| Recruitment | N/A |
| Ethics oversight | N/A |

Note that full information on the approval of the study protocol must also be provided in the manuscript.

# Field-specific reporting

Please select the one below that is the best fit for your research. If you are not sure, read the appropriate sections before making your selection.

☒ Life sciences  ☐ Behavioural & social sciences  ☐ Ecological, evolutionary & environmental sciences

For a reference copy of the document with all sections, see nature.com/documents/nr-reporting-summary-flat.pdf

# Life sciences study design

All studies must disclose on these points even when the disclosure is negative.

| | |
|---|---|
| Sample size | The mono-exponential fits (3 parameters; 2 on normalized data) were performed on a minimum of 10 datapoints when limited by instrumental settings. It was appropriate because we worked with high signal-to-noise ratios for these conditions. We used a minimum of 5 points for affine curve fits (limitation due to significantly long experimental time or heavy sample preparation). |
| Data exclusions | Datapoints were removed when the fluorescence signal-to-noise ratio was so low that the monoexponential fit diverged or gave values outside from the confidence interval predicted by a fit on the datapoints at high signal level. This allowed to report the lower bound for the light intensity that can be calibrated with our instruments and sample conditions. |
| Replication | We performed multiple repeats of the experiments across 3 years. (the number of repeats has been precised in the Main Text). All attempts at replication were successful. The blue-light calibration method was transfered on 7 instruments (not all reported). |

| Randomization | The variety of samples used in this work has necessitated to combine multiple individual expertises. Hence, each experimentalist prepared her/his own samples and made the corresponding experiments. However, all data and their processing have been controled by the corresponding authors. |
| Blinding | The variety of samples used in this work has necessitated to combine multiple individual expertises. Hence, each experimentalist prepared her/his own samples and made the corresponding experiments. However, all data and their processing have been controled by the corresponding authors. Moreover, several measurements have subsequently been reproduced by several experimentalists in different laboratories. |

# Behavioural & social sciences study design

All studies must disclose on these points even when the disclosure is negative.

| Study description | *Briefly describe the study type including whether data are quantitative, qualitative, or mixed-methods (e.g. qualitative cross-sectional, quantitative experimental, mixed-methods case study).* |
| Research sample | *State the research sample (e.g. Harvard university undergraduates, villagers in rural India) and provide relevant demographic information (e.g. age, sex) and indicate whether the sample is representative. Provide a rationale for the study sample chosen. For studies involving existing datasets, please describe the dataset and source.* |
| Sampling strategy | *Describe the sampling procedure (e.g. random, snowball, stratified, convenience). Describe the statistical methods that were used to predetermine sample size OR if no sample-size calculation was performed, describe how sample sizes were chosen and provide a rationale for why these sample sizes are sufficient. For qualitative data, please indicate whether data saturation was considered, and what criteria were used to decide that no further sampling was needed.* |
| Data collection | *Provide details about the data collection procedure, including the instruments or devices used to record the data (e.g. pen and paper, computer, eye tracker, video or audio equipment) whether anyone was present besides the participant(s) and the researcher, and whether the researcher was blind to experimental condition and/or the study hypothesis during data collection.* |
| Timing | *Indicate the start and stop dates of data collection. If there is a gap between collection periods, state the dates for each sample cohort.* |
| Data exclusions | *If no data were excluded from the analyses, state so OR if data were excluded, provide the exact number of exclusions and the rationale behind them, indicating whether exclusion criteria were pre-established.* |
| Non-participation | *State how many participants dropped out/declined participation and the reason(s) given OR provide response rate OR state that no participants dropped out/declined participation.* |
| Randomization | *If participants were not allocated into experimental groups, state so OR describe how participants were allocated to groups, and if allocation was not random, describe how covariates were controlled.* |

# Ecological, evolutionary & environmental sciences study design

All studies must disclose on these points even when the disclosure is negative.

| Study description | *Briefly describe the study. For quantitative data include treatment factors and interactions, design structure (e.g. factorial, nested, hierarchical), nature and number of experimental units and replicates.* |
| Research sample | *Describe the research sample (e.g. a group of tagged Passer domesticus, all Stenocereus thurberi within Organ Pipe Cactus National Monument), and provide a rationale for the sample choice. When relevant, describe the organism taxa, source, sex, age range and any manipulations. State what population the sample is meant to represent when applicable. For studies involving existing datasets, describe the data and its source.* |
| Sampling strategy | *Note the sampling procedure. Describe the statistical methods that were used to predetermine sample size OR if no sample-size calculation was performed, describe how sample sizes were chosen and provide a rationale for why these sample sizes are sufficient.* |
| Data collection | *Describe the data collection procedure, including who recorded the data and how.* |
| Timing and spatial scale | *Indicate the start and stop dates of data collection, noting the frequency and periodicity of sampling and providing a rationale for these choices. If there is a gap between collection periods, state the dates for each sample cohort. Specify the spatial scale from which the data are taken* |
| Data exclusions | *If no data were excluded from the analyses, state so OR if data were excluded, describe the exclusions and the rationale behind them, indicating whether exclusion criteria were pre-established.* |
| Reproducibility | *Describe the measures taken to verify the reproducibility of experimental findings. For each experiment, note whether any attempts to repeat the experiment failed OR state that all attempts to repeat the experiment were successful.* |

| Randomization | *Describe how samples/organisms/participants were allocated into groups. If allocation was not random, describe how covariates were controlled. If this is not relevant to your study, explain why.* |
|---|---|
| Blinding | *Describe the extent of blinding used during data acquisition and analysis. If blinding was not possible, describe why OR explain why blinding was not relevant to your study.* |

Did the study involve field work? ☐ Yes ☐ No

## Field work, collection and transport

| Field conditions | *Describe the study conditions for field work, providing relevant parameters (e.g. temperature, rainfall).* |
|---|---|
| Location | *State the location of the sampling or experiment, providing relevant parameters (e.g. latitude and longitude, elevation, water depth).* |
| Access & import/export | *Describe the efforts you have made to access habitats and to collect and import/export your samples in a responsible manner and in compliance with local, national and international laws, noting any permits that were obtained (give the name of the issuing authority, the date of issue, and any identifying information).* |
| Disturbance | *Describe any disturbance caused by the study and how it was minimized.* |

# Reporting for specific materials, systems and methods

We require information from authors about some types of materials, experimental systems and methods used in many studies. Here, indicate whether each material, system or method listed is relevant to your study. If you are not sure if a list item applies to your research, read the appropriate section before selecting a response.

### Materials & experimental systems

| n/a | Involved in the study |
|---|---|
| ☒ | ☐ Antibodies |
| ☐ | ☒ Eukaryotic cell lines |
| ☒ | ☐ Palaeontology and archaeology |
| ☐ | ☒ Animals and other organisms |
| ☒ | ☐ Clinical data |
| ☒ | ☐ Dual use research of concern |

### Methods

| n/a | Involved in the study |
|---|---|
| ☒ | ☐ ChIP-seq |
| ☒ | ☐ Flow cytometry |
| ☒ | ☐ MRI-based neuroimaging |

## Antibodies

| Antibodies used | *Describe all antibodies used in the study; as applicable, provide supplier name, catalog number, clone name, and lot number.* |
|---|---|
| Validation | *Describe the validation of each primary antibody for the species and application, noting any validation statements on the manufacturer's website, relevant citations, antibody profiles in online databases, or data provided in the manuscript.* |

## Eukaryotic cell lines

Policy information about cell lines and Sex and Gender in Research

| Cell line source(s) | U-2 OS from ATCC cell line HTB-96 lot 64048673 (https://www.atcc.org/products/htb-96)<br>Mycoplasma contamination : Not detected by ATCC |
|---|---|
| Authentication | None of the cell-lines used were authenticated |
| Mycoplasma contamination | Absence of mycoplasma regularly checked internally according to the protocol of the publi : doi.org/10.1038/nprot.2010.43 |
| Commonly misidentified lines<br>(See ICLAC register) | No commonly misidentified cell lines were used in the study. |

# Palaeontology and Archaeology

**Specimen provenance**
*Provide provenance information for specimens and describe permits that were obtained for the work (including the name of the issuing authority, the date of issue, and any identifying information). Permits should encompass collection and, where applicable, export.*

**Specimen deposition**
*Indicate where the specimens have been deposited to permit free access by other researchers.*

**Dating methods**
*If new dates are provided, describe how they were obtained (e.g. collection, storage, sample pretreatment and measurement), where they were obtained (i.e. lab name), the calibration program and the protocol for quality assurance OR state that no new dates are provided.*

☐ Tick this box to confirm that the raw and calibrated dates are available in the paper or in Supplementary Information.

**Ethics oversight**
*Identify the organization(s) that approved or provided guidance on the study protocol, OR state that no ethical approval or guidance was required and explain why not.*

Note that full information on the approval of the study protocol must also be provided in the manuscript.

# Animals and other research organisms

Policy information about studies involving animals; ARRIVE guidelines recommended for reporting animal research, and Sex and Gender in Research

**Laboratory animals**
Chlamydomonas reinhardtii WT CC_124 and WT WT4

**Wild animals**
The study did not involve wild animals

**Reporting on sex**
*Indicate if findings apply to only one sex; describe whether sex was considered in study design, methods used for assigning sex. Provide data disaggregated for sex where this information has been collected in the source data as appropriate; provide overall numbers in this Reporting Summary. Please state if this information has not been collected. Report sex-based analyses where performed, justify reasons for lack of sex-based analysis.*

**Field-collected samples**
The study did not involve samples collected from the field

**Ethics oversight**
No ethical approval is required for microalgae

Note that full information on the approval of the study protocol must also be provided in the manuscript.

# Clinical data

Policy information about clinical studies
All manuscripts should comply with the ICMJE guidelines for publication of clinical research and a completed CONSORT checklist must be included with all submissions.

**Clinical trial registration**
*Provide the trial registration number from ClinicalTrials.gov or an equivalent agency.*

**Study protocol**
*Note where the full trial protocol can be accessed OR if not available, explain why.*

**Data collection**
*Describe the settings and locales of data collection, noting the time periods of recruitment and data collection.*

**Outcomes**
*Describe how you pre-defined primary and secondary outcome measures and how you assessed these measures.*

# Dual use research of concern

Policy information about dual use research of concern

## Hazards

Could the accidental, deliberate or reckless misuse of agents or technologies generated in the work, or the application of information presented in the manuscript, pose a threat to:

No | Yes
- ☐ ☐ Public health
- ☐ ☐ National security
- ☐ ☐ Crops and/or livestock
- ☐ ☐ Ecosystems
- ☐ ☐ Any other significant area

## Experiments of concern

Does the work involve any of these experiments of concern:

No | Yes
- ☐ ☐ Demonstrate how to render a vaccine ineffective
- ☐ ☐ Confer resistance to therapeutically useful antibiotics or antiviral agents
- ☐ ☐ Enhance the virulence of a pathogen or render a nonpathogen virulent
- ☐ ☐ Increase transmissibility of a pathogen
- ☐ ☐ Alter the host range of a pathogen
- ☐ ☐ Enable evasion of diagnostic/detection modalities
- ☐ ☐ Enable the weaponization of a biological agent or toxin
- ☐ ☐ Any other potentially harmful combination of experiments and agents

# ChIP-seq

## Data deposition

☐ Confirm that both raw and final processed data have been deposited in a public database such as GEO.

☐ Confirm that you have deposited or provided access to graph files (e.g. BED files) for the called peaks.

| | |
|---|---|
| **Data access links** *May remain private before publication.* | *For "Initial submission" or "Revised version" documents, provide reviewer access links. For your "Final submission" document, provide a link to the deposited data.* |
| **Files in database submission** | *Provide a list of all files available in the database submission.* |
| **Genome browser session** (e.g. UCSC) | *Provide a link to an anonymized genome browser session for "Initial submission" and "Revised version" documents only, to enable peer review. Write "no longer applicable" for "Final submission" documents.* |

## Methodology

| | |
|---|---|
| **Replicates** | *Describe the experimental replicates, specifying number, type and replicate agreement.* |
| **Sequencing depth** | *Describe the sequencing depth for each experiment, providing the total number of reads, uniquely mapped reads, length of reads and whether they were paired- or single-end.* |
| **Antibodies** | *Describe the antibodies used for the ChIP-seq experiments; as applicable, provide supplier name, catalog number, clone name, and lot number.* |
| **Peak calling parameters** | *Specify the command line program and parameters used for read mapping and peak calling, including the ChIP, control and index files used.* |
| **Data quality** | *Describe the methods used to ensure data quality in full detail, including how many peaks are at FDR 5% and above 5-fold enrichment.* |
| **Software** | *Describe the software used to collect and analyze the ChIP-seq data. For custom code that has been deposited into a community repository, provide accession details.* |

# Flow Cytometry

## Plots

Confirm that:

☐ The axis labels state the marker and fluorochrome used (e.g. CD4-FITC).

☐ The axis scales are clearly visible. Include numbers along axes only for bottom left plot of group (a 'group' is an analysis of identical markers).

☐ All plots are contour plots with outliers or pseudocolor plots.

☐ A numerical value for number of cells or percentage (with statistics) is provided.

## Methodology

| | |
|---|---|
| Sample preparation | *Describe the sample preparation, detailing the biological source of the cells and any tissue processing steps used.* |
| Instrument | *Identify the instrument used for data collection, specifying make and model number.* |
| Software | *Describe the software used to collect and analyze the flow cytometry data. For custom code that has been deposited into a community repository, provide accession details.* |
| Cell population abundance | *Describe the abundance of the relevant cell populations within post-sort fractions, providing details on the purity of the samples and how it was determined.* |
| Gating strategy | *Describe the gating strategy used for all relevant experiments, specifying the preliminary FSC/SSC gates of the starting cell population, indicating where boundaries between "positive" and "negative" staining cell populations are defined.* |

☐ Tick this box to confirm that a figure exemplifying the gating strategy is provided in the Supplementary Information.

# Magnetic resonance imaging

## Experimental design

| | |
|---|---|
| Design type | *Indicate task or resting state; event-related or block design.* |
| Design specifications | *Specify the number of blocks, trials or experimental units per session and/or subject, and specify the length of each trial or block (if trials are blocked) and interval between trials.* |
| Behavioral performance measures | *State number and/or type of variables recorded (e.g. correct button press, response time) and what statistics were used to establish that the subjects were performing the task as expected (e.g. mean, range, and/or standard deviation across subjects).* |

## Acquisition

| | |
|---|---|
| Imaging type(s) | *Specify: functional, structural, diffusion, perfusion.* |
| Field strength | *Specify in Tesla* |
| Sequence & imaging parameters | *Specify the pulse sequence type (gradient echo, spin echo, etc.), imaging type (EPI, spiral, etc.), field of view, matrix size, slice thickness, orientation and TE/TR/flip angle.* |
| Area of acquisition | *State whether a whole brain scan was used OR define the area of acquisition, describing how the region was determined.* |

Diffusion MRI ☐ Used ☐ Not used

## Preprocessing

| | |
|---|---|
| Preprocessing software | *Provide detail on software version and revision number and on specific parameters (model/functions, brain extraction, segmentation, smoothing kernel size, etc.).* |
| Normalization | *If data were normalized/standardized, describe the approach(es): specify linear or non-linear and define image types used for transformation OR indicate that data were not normalized and explain rationale for lack of normalization.* |
| Normalization template | *Describe the template used for normalization/transformation, specifying subject space or group standardized space (e.g. original Talairach, MNI305, ICBM152) OR indicate that the data were not normalized.* |
| Noise and artifact removal | *Describe your procedure(s) for artifact and structured noise removal, specifying motion parameters, tissue signals and physiological signals (heart rate, respiration).* |

| Volume censoring | *Define your software and/or method and criteria for volume censoring, and state the extent of such censoring.* |
|---|---|

## Statistical modeling & inference

| Model type and settings | *Specify type (mass univariate, multivariate, RSA, predictive, etc.) and describe essential details of the model at the first and second levels (e.g. fixed, random or mixed effects; drift or auto-correlation).* |
|---|---|
| Effect(s) tested | *Define precise effect in terms of the task or stimulus conditions instead of psychological concepts and indicate whether ANOVA or factorial designs were used.* |

Specify type of analysis: ☐ Whole brain   ☐ ROI-based   ☐ Both

| Statistic type for inference<br>(See Eklund et al. 2016) | *Specify voxel-wise or cluster-wise and report all relevant parameters for cluster-wise methods.* |
|---|---|
| Correction | *Describe the type of correction and how it is obtained for multiple comparisons (e.g. FWE, FDR, permutation or Monte Carlo).* |

## Models & analysis

| n/a | Involved in the study |
|---|---|
| ☐ | ☐ Functional and/or effective connectivity |
| ☐ | ☐ Graph analysis |
| ☐ | ☐ Multivariate modeling or predictive analysis |

| Functional and/or effective connectivity | *Report the measures of dependence used and the model details (e.g. Pearson correlation, partial correlation, mutual information).* |
|---|---|
| Graph analysis | *Report the dependent variable and connectivity measure, specifying weighted graph or binarized graph, subject- or group-level, and the global and/or node summaries used (e.g. clustering coefficient, efficiency, etc.).* |
| Multivariate modeling and predictive analysis | *Specify independent variables, features extraction and dimension reduction, model, training and evaluation metrics.* |

