## [Peer Review File · Nature Methods]

Peer Review Information

Manuscript Title: Fluorescence to measure light intensity

Corresponding author name(s): Ludovic Jullien

Editorial Notes:

Reviewer Comments & Decisions:

Decision Letter, initial version:
--

Dear Ludovic,

Your Resource, "Fluorescence to measure light intensity", has now been seen by three reviewers. As you will see from their comments below, although the reviewers find your work of considerable potential interest, they have raised a number of concerns. We are interested in the possibility of publishing your paper in Nature Methods, but would like to consider your response to these concerns before we reach a final decision on publication.

We therefore invite you to revise your manuscript to address these concerns. We found the technical concerns to be fairly straightforward, although we think you should spend more time explaining and justifying your choice of dyes and proteins given how possibilities exist (see related comments from refs 1 and 2).

We think the most substantive changes may come in the presentation of the paper. We ask that you revise with a biologist reader in mind, paying careful attention to defining jargon, and making sure that readers come away with a clear sense of how to implement the protocols in their own lab and achieve accurate results. We think this will be aided by the addition of functioning software tools and detailed Supplementary Protocols.

<https://mts-nmeth.nature.com/cgi-bin/main.plex?el=A2M7Xf3A2kvR3J6A9ftdiBSOgvcRVCwDckPwE9egZ>

We hope to receive your revised paper within three months. If you cannot send it within this time, please let us know. In this event, we will still be happy to reconsider your paper at a later date so long as nothing similar has been accepted for publication at Nature Methods or published elsewhere.

OPEN SCIENCE REQUIREMENTS

REPORTING SUMMARY AND EDITORIAL POLICY CHECKLISTS

Please note that these forms are dynamic ‘smart pdfs’ and must therefore be downloaded and completed in Adobe Reader. We will then flatten them for ease of use by the reviewers. If you would like to reference the guidance text as you complete the template, please access these flattened versions at <http://www.nature.com/authors/policies/availability.html>.

IMAGE INTEGRITY

DATA AVAILABILITY

All novel DNA and RNA sequencing data, protein sequences, genetic polymorphisms, linked genotype and phenotype data, gene expression data, macromolecular structures, and proteomics data must be deposited in a publicly accessible database, and accession codes and associated hyperlinks must be provided in the “Data Availability” section.

Please include a “Data availability” subsection in the Online Methods. This section should inform readers about the availability of the data used to support the conclusions of your study, including accession codes to public repositories, references to source data that may be published alongside the paper, unique identifiers such as URLs to data repository entries, or data set DOIs, and any other statement about data availability. At a minimum, you should include the following statement: “The data that support the findings of this study are available from the corresponding author upon request”, describing which data is available upon request and mentioning any restrictions on availability. If DOIs are provided, please include these in the Reference list (authors, title, publisher (repository name), identifier, year). For more guidance on how to write this section please see: <http://www.nature.com/authors/policies/data/data-availability-statements-data-citations.pdf>

CODE AVAILABILITY

Please include a “Code Availability” subsection in the Online Methods which details how your custom code is made available. Only in rare cases (where code is not central to the main conclusions of the paper) is the statement “available upon request” allowed (and reasons should be specified).

For more information on our code sharing policy and requirements, please see: <https://www.nature.com/nature-research/editorial-policies/reporting-standards#availability-of-computer-code>

MATERIALS AVAILABILITY

SUPPLEMENTARY PROTOCOL

To help facilitate reproducibility and uptake of your method, we ask you to prepare a step-by-step Supplementary Protocol for the method described in this paper. We [encourage authors to share their step-by-step experimental protocols](https://www.nature.com/nature-research/editorial-policies/reporting-standards#protocols) on a protocol sharing platform of their choice and report the protocol DOI in the reference list. Nature Portfolio's Protocol Exchange is a free-to-use and open resource for protocols; protocols deposited in Protocol Exchange are citable and can be linked from the published article. More details can found at www.nature.com/protocolexchange/about.

ORCID

Sincerely,
Rita

Rita Strack, Ph.D.
Senior Editor
Nature Methods

Reviewers' Comments:

Reviewer #1:

Remarks to the Author:

A. The article provides two detailed protocols, which exploit fluorescence to enable the retrieval of the spatial distribution of the excitation light intensity for different fluorescence methods, especially fluorescence microscopy, in different commonly used spectral regions. One protocol utilizes the monoexponential fluorescence intensity decays/rises of fluorescent actinometers absorbing and emitting in different wavelength regions. The other protocol exploits an absorbing chemodosimeter in combination with a common photostable fluorophore for the back-calculation of the spatial distribution of the excitation light intensity. The potential for these methods for the calibration of fluorescence imaging systems is subsequently demonstrated.

B. Both methods rely on previously described chemodosimeters (literature cited in the article). This is absolutely mandatory for both methods as the intention is here to provide the community of fluorescence microscopists with available (or at least reproducible) tools for the determination of this very important parameter of fluorescence imaging systems. This approach is original and very significant, given the currently ongoing activities on improving the comparability and quality of fluorescence microscopy data driven by the international network QuaRep.

C. The approach is valid and the quality of the data and the presentation are mostly very good. Here is some (not much!) place for improvement as addressed in section F.

D. Generally, the use of statistics and treatment of uncertainties is well done. What could be helpful for the broad audience of fluorescence microscopists are some general estimates of achievable measurement uncertainties like "with this approach an uncertainty of xxx% can be achieved". If I had to guess, the uncertainty is in the order of 15-20% which is fairly good given the relative ease of both methods and the tools.

E. The conclusions drawn are well supported by the data and experiments done and meet the high standards regarding robustness, validity, and reliability. Suggested minor improvements are addressed in section F.

F. Suggested improvements

i.) To ease the reproducibility of the approach, I suggest to provide HPLC data for the commercial dyes to characterize their purity. This enables other users of these approaches to make sure that the dyes they have at hand are really comparable.

ii.) This approach relies on the assumption of excitation wavelength independent fluorescence quantum yields of the chemodosimeters or the dye rhodamine B. Although I believe that this criterion is met by the suggested tools, this prerequisite should be explicitly addressed. As proof the authors should provide graphs comparing the normalized fluorescence excitation spectra of the fluorophores (equaling the absorption spectra of the emissive species) and the absorption spectra (wavelength-dependent absorption factors, as the fluorescence intensity is proportional to absorption and not absorbance). This could be also included for one dye in Figure 1, panel d.

iii.) The choice of rhodamine B is a bit problematic. This dye is well known for its temperature-dependent fluorescence quantum yield which could possibly cause problems here as thereby, an increase in temperature automatically results in a reduction in fluorescence intensity. Therefore, this dye property should be explicitly addressed with one or two references and it must be checked whether and to which extent this temperature dependence can affect the intensity measurement. Dyes like rhodamine 6G and rhodamine 101 would have been a better choice here as these dyes have a temperature-independent fluorescence quantum yield as the fluorescence quenching channel, the dialkyl amino group is bridged and sterically hindered from rotation (use of julolidine groups)

G. The references are appropriate.

H. The MS is very well written and the abstract and conclusion sections are appropriate. What could be helpful for the general understanding of this approach is a very brief description of which parameters affect a fluorescence signal from the sample and the instrument side (including one or two references, i.e., of IUPAC recommendations) and an explanation of the meaning of fluorescence excitation spectra.

Reviewer #2:

Remarks to the Author:

This manuscript describes two methods and several dyes/fluorescent proteins that can be used to calibrate fluorescence excitation and field uniformity in terms of irradiance. This is not an easy task and this work provides a solution with a manuscript that is very thorough and in depth. The authors present dyes that cover the entire relevant wavelengths for many different applications from the UV to the near

IR. Methods are detailed and there are significant supplemental materials. They demonstrate that results are consistent with measurements using traditional methods such as power meters or spectrometers.

The manuscript is very detailed, supplemental materials give in depth information about dye synthesis, purification of Dronpa, sample preparation and dye characterization. The figures present a lot of information. The characterization of the LED light source with DDAO shown in Figure 5 is particularly impressive. The figure captions contained a lot of details but the formatting was very difficult to follow. I suggest moving some of the detail as text directly on the figure panels themselves or as tables that are easier to follow. Important sample preparation information was buried in the extensive supplemental materials section. For example, the fact that some of the dyes must be dissolved in acetonitrile and require a completely glass sample holder and many dye solutions are only stable for a short time period.

While this manuscript shows a large body of work and broad application it is very technical in nature and will not be of interest to a broad readership in its current format. A more focused manuscript specifically geared towards biological imaging with detailed easy to follow workflows and protocols would be of more broad interest. The authors mention a software tool “ploty data application” and python scripts but these are only mentioned in the discussion and are not described in detail in the manuscript. In addition, it is mentioned that end users can implement the protocols in 1 hour but clear workflows and protocols are not presented. For example, can the biologists enter key parameters about the dye used and the instrument settings that would automatically calculate irradiance from images? Having many dyes covering from UV to near IR is impressive but the fact that the samples all have to be prepared and characterized individually and many are not stable over the long term is problematic.

The manuscript uses a lot of terms that are not common to biologists making it difficult to follow. These terms should be clearly defined. The manuscript is not written with enough information to know exactly what would need to be done for each experiment. Often partial information is provided and the supplemental materials are referred to for key details. Some samples require specialized holders and diffusion of dyes lead to error and the authors claim estimates of irradiance are within an order of magnitude which is not sufficient.

Since Dronpa works so well could a single solution with multiple fluorescent proteins be used to cover the spectral range instead of using different dyes and different sample preparations? Could the proteins/dyes be immobilized on the glass slide to remove diffusion artifacts for calibration of intensity and field uniformity for experiments with thin samples such as cultured cells? In turn, the dyes/fluorescent proteins could be immobilized in a gel (as mentioned at the end of the discussion) with similar refractive index as biological tissue to mimic thicker samples and calibrate irradiance with depth. It would be good to see this kind of data in the manuscript.

Finally, it is not clear if the end user has to characterize the dyes when they use them on their systems of if the characterization that was done in this manuscript can be used to gain key dye parameters? Then users would just need to prepare the dye/fluorescent protein sample, image the sample and perform analysis to calculate irradiance.

Minor Points:

b and c are mixed up in the Figure 2 caption.

Dyes are mentioned in paragraph two on page 3 but then defined later in paragraph three. Define each dye when it is first mentioned.

Reviewer #3:

Remarks to the Author:

(A) In this manuscript Jullien et al. present a set of cleverly designed and comprehensive methods for the qualitative and quantitative measurement of light using fluorescence - either directly (first protocol) or indirectly (second protocol). They show that their approach is applicable in a very wide spectral range but combining different approaches. They apply their methods to wide-field and confocal microscopy - even deep within samples - and demonstrate their methods' use for the wavelength-resolved measurement of light intensity.

(B) This topic is of very high interest for a number of communities both in the photochemistry and in the microscopy field and falls in a very "exciting" time of recent photochemical developments over a wide spectral range which are just waiting to be applied properly for example in microscopy setups. Indeed, local (!) quantification of photons has always been an underinvestigated issue in this respect.

(C) The storyline is extremely succinct - but due the the succinctness also requires quite some background knowledge for the broader understanding of the implications. The arguments for the usefulness are very clearly stated and I support those arguments. The data appear to be of high quality to me - with few exceptions: What is the explanation for the unusual shape of the curve in Figure 1d?

(D) I do not see anything to complain here

(E) I do not see anything to complain here

(F1) I strongly suggest that - where possible - the photoreaction is depicted in the graphics of the main manuscript. This will make it much easier to follow the different photoreaction concepts compared to only describing it with words as it is done now. Also, the structure of DDAO should be shown in the main text.

(F2) Is the fluorescence quantum yield of DDAO the same over the given spectrum? See comment about the unusual curve shape above.

(F3) Many of the underlying principles become only clear when reading the second part of the supporting information. At least the underlying theory of fitting photochemical conversions with simple exponential functions and the meaning of sigma and tau should be briefly discussed in the main text as too many colleagues in the field of photochemistry bother too little about the correct mathematical treatment. True line shapes can be much more complicated with competing chromophores, more than one reaction path and large spectral shifts in the photoprocess. Have the authors confirmed - where possible - that they are dealing with only one, clean photochemical process?

(F4) Typos: p3 "can travel to much of the whole irradiated area" should probably read "can travel too much out of the whole irradiated area"? p4 "As demonstrated in ->the<- Supporting Information" Figure 1 only fits to the text if the labels "b" and "c" are exchanged

(G) Nothing to complain about here.

(H) I especially liked how the relevance and applicability for each of the methods is explicitly stated along with references. The text is rather short but very clear to read. I am wondering if it might be too short for the many different target communities I can imagine to immediately understand the high value of these methods.

Author Rebuttal to Initial comments

Response to referees

Reviewer #1:

A. The article provides two detailed protocols, which exploit fluorescence to enable the retrieval of the spatial distribution of the excitation light intensity for different fluorescence methods, especially fluorescence microscopy, in different commonly used spectral regions. One protocol utilizes the monoexponential fluorescence intensity decays/rises of fluorescent actinometers absorbing and emitting in different wavelength regions. The other protocol exploits an absorbing chemodosimeter in combination with a common photostable fluorophore for the back-calculation of the spatial distribution of the excitation light intensity. The potential for these methods for the calibration of fluorescence imaging systems is subsequently demonstrated.

B. Both methods rely on previously described chemodosimeters (literature cited in the article). This is absolutely mandatory for both methods as the intention is here to provide the community of fluorescence microscopists with available (or at least reproducible) tools for the determination of this very important parameter of fluorescence imaging systems. This approach is original and very significant, given the currently ongoing activities on improving the comparability and quality of fluorescence microscopy data driven by the international network QuaRep.

We thank the reviewer for her/his comments.

C. The approach is valid and the quality of the data and the presentation are mostly very good. Here is some (not much!) place for improvement as addressed in section F.

We thank the reviewer for her/his comments and address her/his concerns in section F below.

D. Generally, the use of statistics and treatment of uncertainties is well done. What could be helpful for the broad audience of fluorescence microscopists are some general estimates of achievable measurement uncertainties like "with this approach an uncertainty of xxx% can be achieved". If I had to guess, the uncertainty is in the order of 15-20% which is fairly good given the relative ease of both methods and the tools.

We addressed the demand of the reviewer by providing: (i) a general estimate of the achievable measurement uncertainty for both general protocols reported in the Main Text. Except for the photosynthetic apparatus of algae (denoted PA in the manuscript), 20 % has been estimated for both the first and second protocols; (ii) specific estimates of the achievable measurement uncertainty for all the fluorescent actinometers reported in the Supporting Information: 20, 20, 20, 20, and 70 % for Cin, Nit, Dronpa-2, DASA, and PA respectively.

E. The conclusions drawn are well supported by the data and experiments done and meet the high standards regarding robustness, validity, and reliability. Suggested minor improvements are addressed in section F.

We thank the reviewer for her/his comments and address her/his concerns in section F below.

F. Suggested improvements

i.) To ease the reproducibility of the approach, I suggest to provide HPLC data for the commercial dyes to characterize their purity. This enables other users of these approaches to make sure that the dyes they have at hand are really comparable.

The revised version of our manuscript reports on the HPLC analysis of both commercially available dyes Rhodamine B and DDAO. We provide information about the HPLC instrument, the separation column, eluting solvents, and retention time in order to facilitate analysis by end-users. We also added the absorption spectrum

and the excitation and emission fluorescence spectra of those dyes to further facilitate the characterization of the dyes that end-users will have at their hand.

ii.) This approach relies on the assumption of excitation wavelength independent fluorescence quantum yields of the chemodosimeters or the dye rhodamine B. Although I believe that this criterium is met by the suggested tools, this prerequisite should be explicitly addressed. As proof the authors should provide graphs comparing the normalized fluorescence excitation spectra of the fluorophores (equaling the absorption spectra of the emissive species) and the absorption spectra (wavelength-dependent absorption factors, as the fluorescence intensity is proportional to absorption and not absorbance). This could be also included for one dye in Figure 1, panel d.

In our first protocol for measuring light intensity with fluorescent actinometers, it was essential to investigate the dependence of the quantum yield of photoconversion on the excitation wavelength since the latter intervenes in Eqs. (S25,S62) of the Supporting Information, which converts the relaxation time τ retrieved from processing the time evolution of the fluorescence signal into the intensity of excitation light. In contrast, as shown in Eqs. (S25,S62) of the Supporting Information, the quantum yield of fluorescence does not enter into the expression of the relaxation time τ . Hence, any possible variation of its value over the range of wavelengths of its light absorption is not detrimental for properly measuring light intensity as long as the wavelength of light excitation remains constant over the measurement. This point has been clarified in the Discussion section of the Main Text of the revised manuscript.

This is different with the second protocol, in which retrieval of light intensity exploits the fluorescence signals measured at two excitation wavelengths. Hence the reviewer is right to ask us to validate that the Kasha's rule is valid for DDAO, which we propose as reporting fluorophore. Hence, we introduced the excitation spectrum of DDAO in the revised manuscript and checked that the absorption and excitation spectra are similar over its relevant range of light absorption. Accordingly, we added a paragraph in subsection D.6 of the Supporting Information related to the validation of DDAO as a reporting fluorophore and included both the absorption and excitation spectra of DDAO in Figure 1d.

iii.) The choice of rhodamine B is a bit problematic. This dye is well known for its temperature-dependent fluorescence quantum yield which could possibly cause problems here as thereby, an increase in temperature automatically results in a reduction in fluorescence intensity. Therefore, this dye property should be explicitly addressed with one or two references and it must be checked whether and to which extent this temperature dependence can affect the intensity measurement. Dyes like rhodamine 6G and rhodamine 101 would have been a better choice here as these dyes have a temperature-independent fluorescence quantum yield as the fluorescence quenching channel, the dialkyl amino group is bridged and sterically hindered from rotation (use of julolidine groups)

Indeed, Rhodamine B is a non photoconverting fluorophore, which is often used as a fluorescent reporter of temperature in relation to the $\sim 1\% \cdot K^{-1}$ dependence of its quantum yield of fluorescence. As discussed above in (ii), the quantum yield of fluorescence does not enter into the expression converting the relaxation time τ retrieved from processing the time evolution of the fluorescence signal into the intensity of excitation light (see Eqs. (S25,S62) of the Supporting Information). Hence, as long as the experiment is performed under isothermal conditions, the temperature dependence of the quantum yield of the fluorescence reporter is not significant for altering the extraction of the light intensity. In fact, this dependence may become only significant in the presence

of strong spatial temperature gradients (in line with the 20% estimate of the achievable measurement uncertainty, typically a spatial gradient of 20 °C over the distance overcome by the reporting fluorophore during the photoconversion time of the fluorescent actinometer). In the Discussion of the Main Text of the revised manuscript, the temperature parameter has now been addressed and its absence of significance is discussed.

G. The references are appropriate.

We thank the reviewer for her/his comments.

H. The MS is very well written and the abstract and conclusion sections are appropriate. What could be helpful for the general understanding of this approach is a very brief description of which parameters affect a fluorescence signal from the sample and the instrument side (including one or two references, i.e., of IUPAC recommendations) and an explanation of the meaning of fluorescence excitation spectra.

We thank the reviewer for her/his comments. The Main Text of the revised manuscript has been enriched with the introduction of a general reference on fluorescence spectroscopy (in the introduction) and a discussion on the significant parameters affecting fluorescence measurements in relation to the sample and to the instrument (in the discussion). We also introduced a glossary explaining various terms of the manuscript (e.g. fluorescence excitation spectrum) in the Supporting Information of the revised manuscript.

Reviewer #2:

This manuscript describes two methods and several dyes/fluorescent proteins that can be used to calibrate fluorescence excitation and field uniformity in terms of irradiance. This is not an easy task and this work provides a solution with a manuscript that is very thorough and in depth. The authors present dyes that cover the entire relevant wavelengths for many different applications from the UV to the near IR. Methods are detailed and there are significant supplemental materials. They demonstrate that results are consistent with measurements using traditional methods such as power meters or spectrometers.

We thank the reviewer for her/his comments.

The manuscript is very detailed, supplemental materials give in depth information about dye synthesis, purification of Dronpa, sample preparation and dye characterization. The figures present a lot of information. The characterization of the LED light source with DDAO shown in Figure 5 is particularly impressive. The figure captions contained a lot of details but the formatting was very difficult to follow. I suggest moving some of the detail as text directly on the figure panels themselves or as tables that are easier to follow. Important sample preparation information was buried in the extensive supplemental materials section. For example, the fact that some of the dyes must be dissolved in acetonitrile and require a completely glass sample holder and many dye solutions are only stable for a short time period.

We thank the reviewer for her/his comments.

As proposed by the reviewer, we shifted some details initially contained in the legends of the Figures of the Main Text towards a specific Table S3 of the first section of the Supporting Information of the revised manuscript.

We were already conscious that it would be favorable to divide the Supporting Information of our manuscript in two parts devoted to end-users and experts respectively. This division has been made even clearer in the revised manuscript, which enabled us to adopt a presentation facilitating the implementation of our protocols for light measurement.

While this manuscript shows a large body of work and broad application, it is very technical in nature and will not be of interest to a broad readership in its current format. A more focused manuscript specifically geared towards biological imaging with detailed easy to follow workflows and protocols would be of more broad interest. The authors mention a software tool “plotly data application” and python scripts but these are only mentioned in the discussion and are not described in detail in the manuscript. In addition, it is mentioned that end users can implement the protocols in 1 hour but clear workflows and protocols are not presented. For example, can the biologists enter key parameters about the dye used and the instrument settings that would automatically calculate irradiance from images? Having many dyes covering from UV to near IR is impressive but the fact that the samples all have to be prepared and characterized individually and many are not stable over the long term is problematic. The article format does not enable us to introduce step-by-step protocols directed towards end users in the Main Text of the revised manuscript. Hence, the Main Text still focuses on the principles underlying our generic protocols for measuring light intensity as well as it reports on the elements aiming at guiding end users for choosing the most relevant one for their own demand. However, we already introduced specific protocols reporting on the implementation of each system in the Supporting Information of the original manuscript in order to facilitate their adoption. This development has been amplified in the Supporting Information of the revised manuscript, which now proposes generic stepwise protocols and workflows, lists of equipment, and troubleshooting, which are further illustrated for each specific system.

The “plotly data application” and python scripts are now mentioned earlier in the Main Text and fully described for their use in the revised manuscript.

To facilitate the adoption of our protocols for measuring light intensity, we introduced a python code building a plotly/Dash app for exploiting the fluorescent actinometers and an Excel sheet for exploiting DDAO to transfer information on light intensity from one wavelength to another one:

- The python code building a plotly/Dash app allows an end user to enter the dye used and the excitation wavelength of the considered light, which generates the relevant sigma value (corresponding to Table 1 of the Main Text). Then the user drags-and-drops a .csv file containing the time evolution of the fluorescence signal from the experiment she/he performed. The code fits a monoexponential to the curve, and uses the extracted characteristic time as well as the sigma value to provide the intensity in $\mu\text{E}\cdot\text{m}^{-2}\cdot\text{s}^{-1}$ or in $\text{W}\cdot\text{m}^{-2}$. An executable file (.exe) is also provided to allow a user not familiar with python to use the interface. When clicking the .exe file, the user should open the address <http://127.0.0.1:8050/> in a web browser and the app should appear as following:

- For the wavelength transfer using DDAO, the Excel file contains the absorbance spectrum of DDAO, which allows the end user to enter: (i) the wavelength $\lambda_{exc,1}$ of the calibrated light and the wavelength $\lambda_{exc,2}$ of the light source to be calibrated; (ii) the fluorescence value of DDAO excited by $\lambda_{exc,1}$: F_1 ; (iii) the fluorescence value of DDAO excited by $\lambda_{exc,2}$: F_2 ; (iv) the previously calibrated intensity I_1 at the wavelength $\lambda_{exc,1}$. Using these inputs, the Excel sheet computes the light intensity I_2 sought for.

We agree with the reviewer that the width of the absorption band of most fluorescent actinometers is limited, which necessitates to have several of them to cover the whole range of wavelengths. This is precisely why we have been interested in introducing (i) microalgae as fluorescent actinometers since they exhibit a broad band light absorption while being easily manipulated as solutions as the other reported fluorescent actinometers; (ii) the second protocol exploiting DDAO as a reporting fluorophore to transfer information on light intensity from one wavelength to another one.

In the revised manuscript, we paid attention to more explicitly explain and justify our choice of dyes and fluorescent proteins given how possibilities exist.

The manuscript uses a lot of terms that are not common to biologists making it difficult to follow. These terms should be clearly defined. The manuscript is not written with enough information to know exactly what would need to be done for each experiment. Often partial information is provided and the supplemental materials are referred to for key details. Some samples require specialized holders and diffusion of dyes lead to error and the authors claim estimates of irradiance are within an order of magnitude which is not sufficient.

To address the reviewer concerns, we introduced in the revised manuscript:

- a glossary explaining various terms of the manuscript (e.g. fluorescence excitation spectrum) in the Supporting Information;

- generic stepwise protocols and workflows, lists of equipment, and troubleshooting, which are illustrated for each specific system. Please note that the article format does not enable us to introduce all these aspects in the Main Text of the manuscript, which explains that they are reported in the Supporting Information.

There is no highly specific holder reported in the manuscript for light measurement. In the Supporting Information of the revised manuscript, we introduced the list of equipment necessitated for light measurement.

The molecular diffusion of dyes (illustrated in Figure 2g-i) reduces the spatial resolution of the map of light intensity of the image. However, it does not introduce any significant deviation modifying the order of magnitude

of the measured light intensity as long as molecular diffusion remains spatially limited at the time scale of the actinometer photoconversion. This point has been made precise in the Main Text of the revised manuscript.

The photosynthetic apparatus of algae (denoted PA in the manuscript) is the only fluorescent actinometer for which the uncertainty on the measurement of light intensity is higher than 20% in view of some sensitivity of its fluorescence response to its physiological state. Yet, it provides a good estimate of light intensity for the entire visible range of wavelengths, which is in fact better than an order of magnitude. Hence, we modified the PA presentation, which is now “we eventually report on the last actinometer, the photosynthetic apparatus of algae (denoted PA), which can provide an estimate of light intensity for the entire visible range of wavelengths” in the Main Text of the revised manuscript.

Since Dronpa works so well could a single solution with multiple fluorescent proteins be used to cover the spectral range instead of using different dyes and different sample preparations? Could the proteins/dyes be immobilized on the glass slide to remove diffusion artifacts for calibration of intensity and field uniformity for experiments with thin samples such as cultured cells? In turn, the dyes/fluorescent proteins could be immobilized in a gel (as mentioned at the end of the discussion) with similar refractive index as biological tissue to mimic thicker samples and calibrate irradiance with depth. It would be good to see this kind of data in the manuscript.

The repertoire of photoswitchable fluorescent proteins is under expansion. However, we still lack some representatives to fill the whole wavelengths of the visible range. In addition, working with a mixture of fluorescent actinometers makes the system to be quite sensitive to the relative proportions of the mixture components, which would complicate the analysis. In fact, to overcome the limitation of the width of the absorption band of most fluorescent actinometers, we presently rather propose to either adopt photosynthetic organisms as fluorescent actinometers (since they exhibit a broad band light absorption while being easily accessible) or use the second protocol exploiting DDAO as a reporting fluorophore to transfer information on light intensity from one wavelength to another one. These points have been made clearer in the Main Text of the revised manuscript.

In principle, all the reported fluorescent actinometers can be immobilized to remove diffusion artifacts for calibration of intensity and field uniformity for experiments with thin samples such as cultured cells. This is precisely what has been illustrated in Figure 2g-i. Key is here to choose the right embedding medium. For the water-soluble fluorescent actinometers, a polyacrylamide gel is a good choice. Here, low mesh size and gel sealing after sandwiching between glass slides ensure both low molecular diffusion and absence of water loss respectively. Immobilization of the synthetic fluorescent actinometers in organic media would require some more developments either by modifying their molecular structure (by adding anchoring groups to graft them on a surface) or by embedding them in a highly viscous (e.g. a polymer) or solid organic matrix. However, one would have then to characterize their photochemical properties in those media, which has not been covered during the present study. The latter points have been addressed in the discussion of the Main Text of the revised manuscript.

The reviewer proposes to immobilize the dyes/fluorescent proteins in a gel with similar refractive index as biological tissue to mimic thicker samples and calibrate irradiance with depth. In fact, that was precisely the purpose of the experiment reported in Figure 3g,h. In this experiment, we used Dronpa-2-labeled bacteria as fluorescent actinometers, which have been immobilized in a light scattering agarose gel. The relevance of this experiment has been made more precise in the discussion of the Main Text of the revised manuscript.

Finally, it is not clear if the end user has to characterize the dyes when they use them on their systems or if the characterization that was done in this manuscript can be used to gain key dye parameters? Then users would just need to prepare the dye/fluorescent protein sample, image the sample and perform analysis to calculate irradiance.

The characterization of the fluorescent actinometers has been done in the manuscript and the end users can exploit the reported parameters to reliably measure light intensity as long as they follow the reported measurement protocols. This point has been made clearer in the Main Text of the revised manuscript.

Minor Points:

b and c are mixed up in the Figure 2 caption.

We thank the reviewer and corrected this point.

Dyes are mentioned in paragraph two on page 3 but then defined later in paragraph three. Define each dye when it is first mentioned.

We revised the introduction of the dyes in order to avoid this drawback.

Reviewer #3:

Remarks to the Author:

(A) In this manuscript Jullien et al. present a set of cleverly designed and comprehensive methods for the qualitative and quantitative measurement of light using fluorescence - either directly (first protocol) or indirectly (second protocol). They show that their approach is applicable in a very wide spectral range but combining different approaches. They apply their methods to wide-field and confocal microscopy - even deep within samples - and demonstrate their methods' use for the wavelength-resolved measurement of light intensity.

We thank the reviewer for her/his comments.

(B) This topic is of very high interest for a number of communities both in the photochemistry and in the microscopy field and falls in a very "exciting" time of recent photochemical developments over a wide spectral range which are just waiting to be applied properly for example in microscopy setups. Indeed, local (!) quantification of photons has always been an underinvestigated issue in this respect.

We thank the reviewer for her/his comments.

(C) The storyline is extremely succinct - but due the succinctness also requires quite some background knowledge for the broader understanding of the implications. The arguments for the usefulness are very clearly stated and I support those arguments. The data appear to be of high quality to me - with few exceptions: What is the explanation for the unusual shape of the curve in Figure 1d?

We thank the reviewer for her/his comments.

Indeed, DDAO exhibits rather singular shapes of both its absorption and fluorescence emission spectra. Although we do not have a clear photophysical explanation, similar spectra have been reported in the literature (see for instance references 36 and 37 of the revised Main Text).

(D) I do not see anything to complain here

(E) I do not see anything to complain here

(F1) I strongly suggest that - where possible - the photoreaction is depicted in the graphics of the main manuscript. This will make it much easier to follow the different photoreaction concepts compared to only describing it with words as it is done now. Also, the structure of DDAO should be shown in the main text.

We understand the concern of the reviewer. However, we lack space to introduce the photoreactions in the Main Text, which has a constrained format. Nevertheless, to address his concern, the Supporting Information of the revised manuscript now contains generic stepwise protocols and workflows, lists of equipment, and troubleshooting, which are illustrated for each specific system. It is in this section that we introduced the photoreaction for each system.

We introduced the structure of DDAO in the Figure 1d of the Main Text of the revised manuscript.

(F2) Is the fluorescence quantum yield of DDAO the same over the given spectrum? See comment about the unusual curve shape above.

The reviewer is right to ask us to validate that the Kasha's rule is valid for DDAO, which we propose as reporting fluorophore. Hence, we introduced the excitation spectrum of DDAO in the revised manuscript and checked that the absorption and excitation spectra are similar over its relevant range of light absorption. Accordingly, we added a paragraph in subsection D.6 of the Supporting Information related to the validation of DDAO as a reporting fluorophore and included both the absorption and excitation spectra of DDAO in Figure 1d.

(F3) Many of the underlying principles become only clear when reading the second part of the supporting information. At least the underlying theory of fitting photochemical conversions with simple exponential functions and the meaning of sigma and tau should be briefly discussed in the main text as too many colleagues in the field of photochemistry bother too little about the correct mathematical treatment. True line shapes can be much more complicated with competing chromophores, more than one reaction path and large spectral shifts in the photoprocess. Have the authors confirmed - where possible - that they are dealing with only one, clean photochemical process?

The underlying theory of fitting photochemical conversions with simple exponential functions and the meaning of sigma and tau are now briefly discussed in the discussion of the Main Text of the revised manuscript.

The mechanisms underlying the photoconversion of photoactive molecules most often involve multiple steps and the mechanistic reduction for generating simple data processing requires thorough attention (see for instance subsection B.2 of the Supporting Information of Chouket et al, *Nat. Comm.* 2022). In fact, the purpose of investigating the light intensity-dependence of the characteristic time τ retrieved from monoexponentially fitting the time evolution of the fluorescence response of the various actinometers reported in the Supporting Information is precisely to support (and also to circumvent) the relevance of monoexponential fitting for processing the time evolution of the fluorescence signal from the reported fluorescent actinometers. It resulted in defining ranges of light intensity in which we ensured that the light-driven step of the photoconversion mechanism is constant and rate limiting so as to make this protocol of data processing reliable.

(F4) Typos: p3 "can travel to much of the whole irradiated area" should probably read "can travel too much out of the whole irradiated area"? p4 "As demonstrated in ->the<- Supporting Information" Figure 1 only fits to the text if the labels "b" and "c" are exchanged

Indeed, the expression was misleading. We replaced "However, if the timescales are such that molecules can travel to much of the whole irradiated area, only mean light intensity values can be obtained" by "However, if the

molecules can visit the whole irradiated area at the timescale of the actinometer photoconversion, only mean light intensity values can be obtained”.

(G) Nothing to complain about here.

(H) I especially liked how the relevance and applicability for each of the methods is explicitly stated along with references. The text is rather short but very clear to read. I am wondering if it might be too short for the many different target communities I can imagine to immediately understand the high value of these methods.

We attempted to at the most explicit in the Main Text of the revised manuscript in order to facilitate understanding by the broadest community of end users (e.g. by introducing a glossary and making them aware of the key parameters securing reliable light measurements).

Decision Letter, first revision:

Dear Ludovic,

Thank you for submitting your revised manuscript "Fluorescence to measure light intensity" (NMETH-A51899A). It has now been seen by the original referees and their comments are below. The reviewers find that the paper has improved in revision, and therefore we'll be happy in principle to publish it in Nature Methods, pending minor revisions to satisfy the referees' final requests and to comply with our editorial and formatting guidelines.

In response to reviewer 2, we ask that you organize and present the material as clearly as possible for the biologist reader (with some microscopy expertise) to implement with as much ease as possible. Please leave the more expert technical details well-organized in the supplement.

We are now performing detailed checks on your paper and will send you a checklist detailing our editorial and formatting requirements in about two weeks. Please do not upload the final materials and make any revisions until you receive this additional information from us.

TRANSPARENT PEER REVIEW

Nature Methods offers a transparent peer review option for new original research manuscripts submitted from 17th February 2021. We encourage increased transparency in peer review by publishing the reviewer comments, author rebuttal letters and editorial decision letters if the authors agree. Such peer review material is made available as a supplementary peer review file. Please state in the cover letter 'I wish to participate in transparent peer review' if you want to opt in, or 'I do not wish to participate in transparent peer review' if you don't. Failure to state your preference will result in delays in accepting your manuscript for publication.

ORCID

IMPORTANT: Non-corresponding authors do not have to link their ORCIDs but are encouraged to do so. Please note that it will not be possible to add/modify ORCIDs at proof. Thus, please let your co-authors

know that if they wish to have their ORCID added to the paper they must follow the procedure described in the following link prior to acceptance:

Sincerely,
Rita

Rita Strack, Ph.D.
Senior Editor
Nature Methods

Reviewer #1 (Remarks to the Author):

Since this is a revision, I can only confirm that all points raised by me and the other reviewers have been excellently considered in the revised MS and SI and have been addressed in the rebuttal in a detailed and clear way.

There is not much to add as to congratulate the authors to this important work.

Reviewer #2 (Remarks to the Author):

The revised manuscript is significantly improved. Thank you for all of the changes.

I still find the figures and captions difficult to follow.

Having to refer to a supplemental table is not ideal.

I understand the space limitations with the current manuscript format but one of the most valuable aspects of the paper is the method of making the measurements and the software. This information is still difficult to find within very long supplemental materials.

This is an excellent quality paper on an important topic that is in depth and thorough. However, it is highly technical and I do not think it will be of general interest to the broad Nature Methods readership.

Reviewer #3 (Remarks to the Author):

I see that Jullien et al. have responded very well not only to my comments but also to the comments of the other reviewers. As for the answers to my comments I am fully satisfied and have no further suggestions or concerns. I also think that the comments of the other reviewers were addressed properly. I believe that this was a "healthy" revision process and am sure that the revised version will be a very fine manuscript in Nature Methods that will receive much attention.

Author Rebuttal, first revision:

Please first find below our point-by-point response to the reviewers:

- **Reviewer #1 (Remarks to the Author):** *Since this is a revision, I can only confirm that all points raised by me and the other reviewers have been excellently considered in the revised MS and SI and have been addressed in the rebuttal in a detailed and clear way. There is not much to add as to congratulate the authors to this important work.*

We thank the reviewer for her/his comments.

- **Reviewer #2 (Remarks to the Author):** *The revised manuscript is significantly improved. Thank you for all of the changes. I still find the figures and captions difficult to follow. Having to refer to a supplemental table is not ideal. I understand the space limitations with the current manuscript format but one of the most valuable aspects of the paper is the method of making the measurements and the software. This information is still difficult to find within very long supplemental materials.*

We have been asked by the Editors to reduce by 10% the number of words in the Main Text. While proceeding, we attempted to present the material of the Main Text of our manuscript as clearly as possible for the biologist reader with some microscopy expertise in order to implement our protocols with as much ease as possible. In particular, the section "Discussion" has been modified with a first paragraph emphasizing on the work flow to be adopted by end users to measure light intensity with our protocols. In particular we added the sentence: "After a selection relying on the wavelength to investigate and access to chemical or biological facilities, end users should implement the protocol illustrated in Figure 1a and detailed in Supplementary Section 2 upon using the downloadable application for data processing. Alternatively, they should use the commercially available DDAO to retrieve the light intensity of their desired light source after calibrating another light source with the actinometer they can access."

This is a excellent quality paper on an important topics that is in depth and thorough. However, it is highly technical and I do not think it will be of general interest to the broad Nature Methods readership.

We thank the reviewer for her/his comments.

*- **Reviewer #3 (Remarks to the Author):** I see that Jullien et al. have responded very well not only to my comments but also to the comments of the other reviewers. As for the answers to my comments I am fully satisfied and have no further suggestions or concerns. I also think that the comments of the other reviewers were addressed properly. I believe that this was a "healthy" revision process and am sure that the revised version will be a very fine manuscript in Nature Methods that will receive much attention.*

We thank the reviewer for her/his comments.

Final Decision Letter:

Dear Ludovic

I am pleased to inform you that your Article, "Fluorescence to measure light intensity", has now been accepted for publication in Nature Methods. Your paper is tentatively scheduled for publication in our December print issue, and will be published online prior to that. The received and accepted dates will be March 6, 2023 and October 2, 2023. This note is intended to let you know what to expect from us over the next month or so, and to let you know where to address any further questions.

Over the next few weeks, your paper will be copyedited to ensure that it conforms to Nature Methods style. Once your paper is typeset, you will receive an email with a link to choose the appropriate publishing options for your paper and our Author Services team will be in touch regarding any additional information that may be required.

You will receive a link to your electronic proof via email with a request to make any corrections within 48 hours. If, when you receive your proof, you cannot meet this deadline, please inform us at rjsproduction@springernature.com immediately.

Please note that *Nature Methods* is a Transformative Journal (TJ). Authors may publish their research with us through the traditional subscription access route or make their paper immediately open access through payment of an article-processing charge (APC). Authors will not be required to make a final decision about access to their article until it has been accepted. [Find out more about Transformative Journals](https://www.springernature.com/gp/open-research/transformative-journals)

Your paper will now be copyedited to ensure that it conforms to Nature Methods style. Once proofs are generated, they will be sent to you electronically and you will be asked to send a corrected version within 24 hours. It is extremely important that you let us know now whether you will be difficult to contact over the next month. If this is the case, we ask that you send us the contact information (email, phone and fax) of someone who will be able to check the proofs and deal with any last-minute problems.

If, when you receive your proof, you cannot meet the deadline, please inform us at rjsproduction@springernature.com immediately.

Once your manuscript is typeset and you have completed the appropriate grant of rights, you will receive a link to your electronic proof via email with a request to make any corrections within 48 hours. If, when you receive your proof, you cannot meet this deadline, please inform us at rjsproduction@springernature.com immediately.

Once your paper has been scheduled for online publication, the Nature press office will be in touch to confirm the details.

Once your paper has been scheduled for online publication, the Nature press office will be in touch to confirm the details.

Content is published online weekly on Mondays and Thursdays, and the embargo is set at 16:00 London time (GMT)/11:00 am US Eastern time (EST) on the day of publication. If you need to know the exact publication date or when the news embargo will be lifted, please contact our press office after you have submitted your proof corrections. Now is the time to inform your Public Relations or Press Office about your paper, as they might be interested in promoting its publication. This will allow them time to prepare an accurate and satisfactory press release. Include your manuscript tracking number NMETH-A51899B and the name of the journal, which they will need when they contact our office.

About one week before your paper is published online, we shall be distributing a press release to news organizations worldwide, which may include details of your work. We are happy for your institution or funding agency to prepare its own press release, but it must mention the embargo date and Nature Methods. Our Press Office will contact you closer to the time of publication, but if you or your Press Office have any inquiries in the meantime, please contact press@nature.com.

Nature Portfolio journals [encourage authors to share their step-by-step experimental protocols](https://www.nature.com/nature-research/editorial-policies/reporting-standards#protocols) on a protocol sharing platform of their choice. Nature Portfolio 's Protocol Exchange is a free-to-use and open resource for protocols; protocols deposited in Protocol Exchange are citable and can be linked from the published article. More details can found at www.nature.com/protocolexchange/about.

Best regards,
Rita

Rita Strack, Ph.D.
Senior Editor
Nature Methods